# The melanocortin action is biased toward protection from weight loss in mice

Hongli Li[1,2,7], Yuanzhong Xu [2,7], Yanyan Jiang[2], Zhiying Jiang[2], Joshua Otiz-Guzman[3], Jessie C. Morrill[2,4], Jing Cai[2,4], Zhengmei Mao[2], Yong Xu [5], Benjamin R. Arenkiel [3], Cheng Huang [1] ✉ & Qingchun Tong [2,4,6] ✉

The melanocortin action is well perceived for its ability to regulate body weight bidirectionally with its gain of function reducing body weight and loss of function promoting obesity. However, this notion cannot explain the difficulty in identifying effective therapeutics toward treating general obesity via activation of the melanocortin action. Here, we provide evidence that altered melanocortin action is only able to cause one-directional obesity development. We demonstrate that chronic inhibition of arcuate neurons expressing proopiomelanocortin (POMC) or paraventricular hypothalamic neurons expressing melanocortin receptor 4 (MC4R) causes massive obesity. However, chronic activation of these neuronal populations failed to reduce body weight. Furthermore, gain of function of the melanocortin action through overexpression of MC4R, POMC or its derived peptides had little effect on obesity prevention or reversal. These results reveal a bias of the melanocortin action towards protection of weight loss and provide a neural basis behind the well-known, but mechanistically ill-defined, predisposition to obesity development.

Body weight homeostasis is controlled by balanced energy intake and expenditure. Body weight gain occurs when energy intake exceeds expenditure and can ultimately lead to obesity. Obesity is a significant contributor to an array of other metabolic diseases, including type 2 diabetes, and is a major risk factor for some cancers[1,2]. Despite extensive research on body weight regulation and continued efforts towards the development of therapeutics for the mitigation of obesity, the prevalence of obesity has reached epidemic levels, with more than one-third of the American population suffering from obesity and its associated life challenges[3,4]. As obesity only develops in a subset of the human population and is closely associated with recent social and economic development, it is generally believed that obesity results from an interaction between individual genetic makeups and social/environmental changes[4,5]. To explain the rapid obesity epidemic, the thrifty gene hypothesis poses that a key subset of "thrifty" genes have been selected throughout evolution during times of nutrient scarcity, thus facilitating positive energy balance in times with nutrient surplus[6]. However, this hypothesis cannot explain why only a portion of the human population develops obesity. In lieu of the thrifty gene hypothesis, a new "drifty" gene hypothesis has been proposed, arguing for genetic drift in the genes encoding components that regulate metabolism and control upper limits on body fatness[7–9]. Nevertheless,

[1]School of Pharmacy, Shanghai University of Traditional Chinese Medicine, 1200 Cailun Road, Shanghai 201203, China. [2]Brown Foundation of Molecular Medicine for the Prevention of Human Diseases of McGovern Medical School, University of Texas Health Science Center at Houston, Houston, TX 77030, USA. [3]Department of Molecular and Human Genetics and Department of Neuroscience, Baylor College of Medicine, and Jan and Dan Duncan Neurological Research Institute, Texas Children's Hospital, Houston, TX, USA. [4]MD Anderson Cancer Center & UTHealth Graduate School for Biomedical Sciences, University of Texas Health Science at Houston, 77030 Houston, TX, USA. [5]Children's Nutrition Research Center, Department of Pediatrics, Baylor College of Medicine, One Baylor Plaza, Houston, TX 77030, USA. [6]Department of Neurobiology and Anatomy of McGovern Medical School, University of Texas Health Science Center at Houston, Houston, TX 77030, USA. [7]These authors contributed equally: Hongli Li, Yuanzhong Xu. ✉e-mail: chuang@shutcm.edu.cn; Qingchun.tong@uth.tmc.edu

the biological underpinning for the observed predisposition to obesity development remains elusive.

Extensive research over the past few decades has identified the hypothalamus as a key regulator of feeding and energy expenditure. In particular, the melanocortin signaling has been identified as a critical and conserved pathway for body weight homeostasis[10]. This pathway consists of proopiomelanocortin (POMC)- and agouti-related protein (AgRP)-expressing neurons in the arcuate hypothalamic nucleus (Arc) and their downstream neurons that express melanocortin receptors, with melanocortin receptor 4 (MC4R)-expressing neurons as a primary target. Arcuate POMC neurons release α-melanocyte releasing hormone (α-MSH), a POMC-derived anorexigenic peptide, and AgRP neurons release orexigenic AgRP, to dynamically and inversely regulate MC4Rs[10].

Compelling genetic studies conducted in both rodents and humans have demonstrated that POMC or MC4R loss of function causes obesity and is associated with increased feeding and reduced energy expenditure[11–15]. Of note, a sizable proportion of the human population with monogenetic obesity have genetic mutations in the melanocortin pathway; MC4R mutations are the most common form of monogenetic obesity[16–20]. Pharmacological activation of MC4Rs with α-MSH or related agonists reduces feeding and facilitates acute body weight loss, while inhibition of MC4Rs with AgRP increases feeding[21–24]. Recent optogenetic and chemogenetic studies also support that changes in the POMC and AgRP neuron activity bidirectionally alter feeding; POMC neurons function in an anorexigenic role, whereas AgRP neurons serve orexigenic functions[25–28]. Physiologically, POMC and AgRP neurons have been suggested to mediate the actions of hormones, including leptin, which simultaneously activates POMC neurons and inhibits AgRP neurons to limit body weight gain in response to increasing leptin levels[29,30]. Together, these observations support that the melanocortin pathway regulates body weight in a bidirectional fashion, whereby increased pathway activities reduce body weight and decreased activities promote weight gain.

Despite overwhelming evidence that supports the bidirectional nature of melanocortin action on body weight regulation, this notion has been challenged in numerous ways. First, the predicted leptin activation of the melanocortin action on body weight reduction contradicts notions of obesity development associated with hyperleptinemia[29,31]. Although the interpretation of these results is complicated by the presence of leptin resistance, there is a debate on whether leptin resistance truly exists[31–33]. Second, given the expected effects of reducing weight gain by the activation of the melanocortin pathway, attempts at developing effective anti-obesity therapeutics based on the activation of MC4Rs remain a significant challenge[34–36]. It is worth noting that one MC4R agonist (Setmelanotide) has been approved for use in patients with obesity, but it can only be prescribed to patients who are confirmed to have obesity-causing gene mutations in the melanocortin pathway instead of general obesity[37]. Third, extensive human genetic studies have identified numerous mutations in POMC, MC4R, and other genes related to the melanocortin pathway in association with body weight imbalance[13–16,38,39]. However, most, if not all, of obesity-associated gene variants are loss of function mutations, whereas there is a lack of reports on mutations with a gain of function associated with reduced body weight. Although a gain of function of MC4R mutation has been suggested to be associated with reduced body weight, recent studies suggest otherwise[40–42]. These inconsistent observations raise an issue on the rationale of using the melanocortin pathway as a therapeutic target against the current obesity epidemic.

In an effort to clarify the bidirectional modulation of the melanocortin action on body weight regulation, we generated animal models with either chronic loss or gain of function in POMC or MC4R neurons. Consistent with the published data from rodents and humans, chronic loss of function of POMC or MC4R neurons results in massive obesity. However, a chronic gain of function in POMC neurons through chronic activation, or overexpression of POMC or its derived peptides, fails to cause an effect on body weight reduction or obesity prevention/reversal. Similar observations were also seen in mice with either chronic activation of PVH MC4R (PVH^MC4R) neurons or targeted overexpression of MC4R. Together, these results alongside our previous findings that chronic activation of AgRP neurons causes massive obesity development, whereas chronic inhibition has no impact on body weight[43], suggest that the melanocortin action plays a limited role in body weight loss in normal or obese conditions and is biased toward protection of weight loss.

## Results

### Chronic inhibition of POMC neurons causes obesity

To examine the impact of chronic inhibition of POMC neurons on body weight regulation, we stereotaxically injected a conditional AAV-Flex-Kir2.1 into the Arc of POMC-Cre mice (Fig. 1a). Kir2.1 expression has been previously shown to reduce neuronal activity in a chronic fashion[43]. Peptide expression is known to be sensitive to neuron activity levels[44]. To examine the effect of Kir2.1 expression on POMC peptides, the virus was selectively delivered to one side of the Arc (Fig. 1a, b). Immunostaining for β-endorphin showed an obvious reduction in expression on the injected side, compared to the control side (arrows in Fig. 1b, right panel), suggesting a strong inhibitory effect on POMC neurons with Kir2.1 expression. To further confirm the targeted inhibition, we made electrophysiological recordings from POMC neurons and found that, compared to controls (Supplementary Fig. 1a), Kir2.1-expressing POMC neurons exhibited a more hyperpolarized resting membrane potential (REM) (Supplementary Fig. 1b, d). In addition, Kir2.1-expressing POMC neurons showed a reduced input resistance (Supplementary Fig. 1e) and lower spontaneous firing frequencies (Supplementary Fig. 1f). Further, compared to controls (Supplementary Fig. 1g, j), Kir2.1-expressing POMC neurons showed a higher rheobase (Supplementary Fig. 1h, j). Together, these observed differences in both in vivo and in vitro conditions confirmed that POMC neurons targeted for Kir2.1 expression showed reduced activity.

To examine the effect of chronic POMC neuron inhibition on body weight, we bilaterally delivered Kir2.1 to the Arc of POMC-Cre mice and then monitored their body weight for 10 weeks (Fig. 1c). In mice with confirmed targeted expression (Fig. 1d), we observed dramatically increased body weights that reached 50 g by 10 weeks following viral delivery (Fig. 1e). The increased weight was largely due to elevated fat mass (Fig. 1f) and a slight increase in lean mass (Fig. 1g), suggesting obesity development. The obesity was contributed to increased food intake (Fig. 1h). When measured prior to any observed changes in body weight (Supplementary Fig. 2a), Kir2.1-expressing mice showed no significant difference in $O_2$ consumption (Fig. 1i and Supplementary Fig. 2b), although they exhibited significantly reduced locomotion levels (Fig. 1j and Supplementary Fig. 2c). Notably, the observed massive obesity in the Kir2.1-injected mice was refractory to icv leptin treatment (Supplementary Fig. 2d, e). The function of the leptin used was independently validated, given that the same treatment effectively reduced obesity in ob/ob mice (Supplementary Fig. 2e) and dramatically increased p-STAT3 expression in the Arc (Supplementary Fig. 2f). In addition, we noted a similar body weight gain in females (Supplementary Fig. 2g). Thus, chronic inhibition of POMC neurons leads to massive obesity in both males and females, which is mainly attributed to hyperphagia.

### Chronic activation of POMC neurons does not reduce body weight

To test the effect of chronic activation of POMC neurons, we expressed a mutated version of the bacterial sodium channel, NachBac[43], via stereotaxic delivery of a conditional AAV-Flex-NachBac-GFP virus to the Arc of POMC-Cre mice. When delivered to one side of the Arc (Fig. 2a),

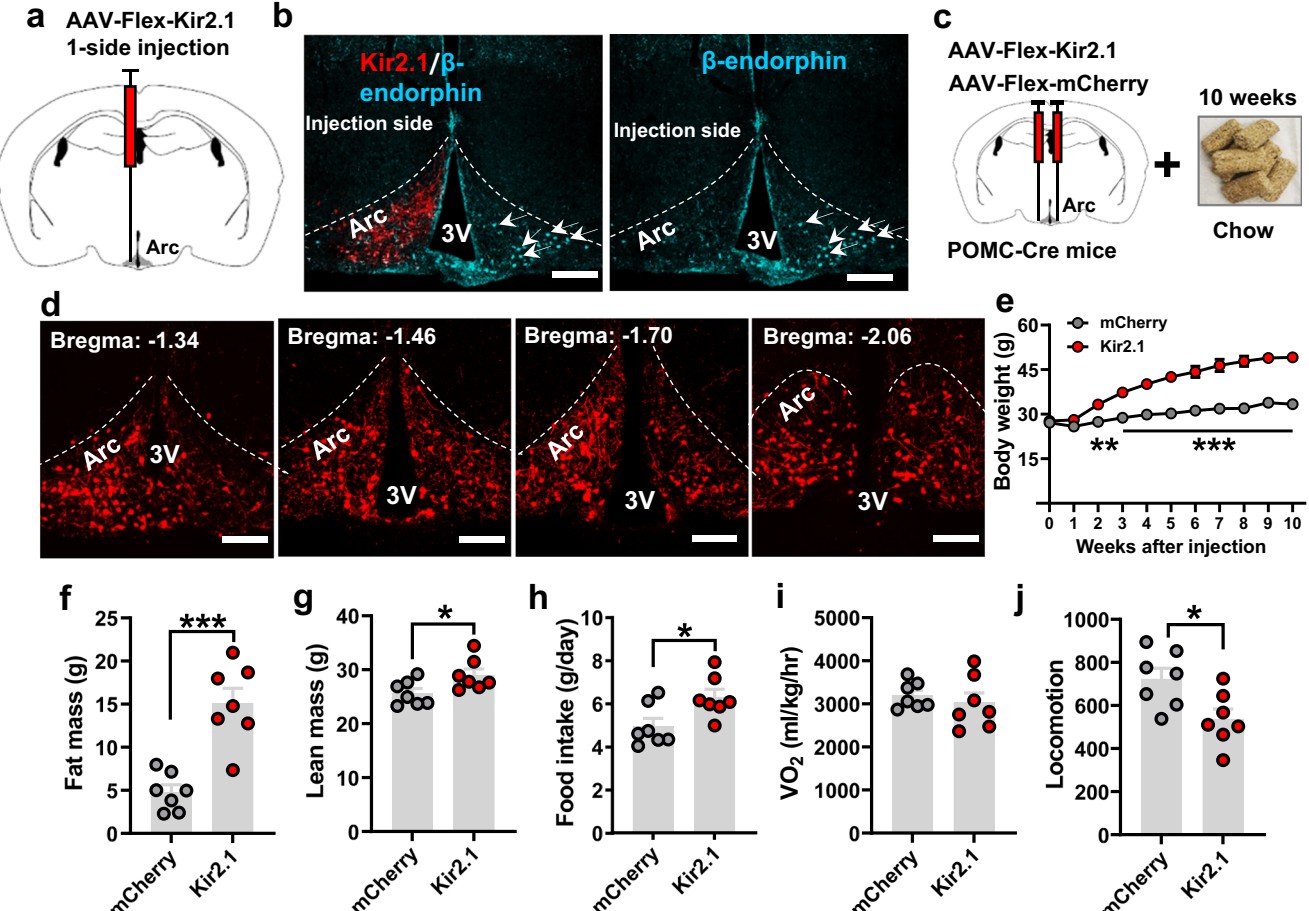

**Fig. 1 | Chronic inhibition of Arc POMC neurons caused obesity. a** Diagram depicting one side injection of AAV-Felx-Kir2.1-mCherry virus to Arc POMC neurons. **b** Expression pattern of viral vector expression (mCherry) and β-endorphin (cyan) or β-endorphin alone to demonstrate a higher level of β-endorphin at the non-injected side (arrows). **c** Diagram depicting bilateral injection of AAV-Flex-Kir2.1-mCheery virus to the Arc of POMC-Cre mice feeding on a chow diet for 10 weeks after viral delivery. **d** Representative expression patterns of the injected virus in the Arc in a rostral to caudal dimension. 3 V: third ventricle, Arc: arcuate nucleus. Scale bars: 50 μm. **e** Weekly body weight of male POMC-Cre mice with bilateral delivery of Kir2.1 or control virus to the Arc ($n$ = 8 mice/each, two-way ANOVA, Control vs. Kir2.1, **$p$ = 0.009 at 2 weeks post viral injection, ***$p$ < 0.001

at 3–10 weeks post viral injection). **f–h** Comparison in fat mass (**f**, $n$ = 7 mice/each, two-tailed unpaired $t$-tests, ***$p$ = 0.0002), lean mass (**g**, $n$ = 7 mice/each, two-tailed unpaired $t$-tests, *$p$ = 0.0334) and feeding (**h**, $n$ = 7 mice/each, two-tailed unpaired $t$-tests, *$p$ = 0.0218) in the mice shown in **e** at 10 weeks after viral delivery. **i, j** Comparison between the groups of mice in $O_2$ consumption (**i**, $n$ = 7 mice/each, two-tailed unpaired $t$-tests, $p$ = 0.5062) and locomotion (**j**, $n$ = 7 mice/each, two-tailed unpaired $t$-tests, *$p$ = 0.0179) measured between 2–3 weeks after viral injection when the body weight difference between the groups was minimal. All data were presented as mean ± SEM. Source data are provided in the Source Data file.

we confirmed selective NachBac expression in POMC neurons (Fig. 2b), and observed much greater numbers of neurons with c-Fos expression in the injected side, compared to the control side (Fig. 2b, c). Similar to previously described in ref. 43, NachBac-expressing POMC neurons displayed altered and prolonged action potentials (Supplementary Fig. 1c), reduced input resistance (Supplementary Fig. 1e), and a lower rheobase (Supplementary Fig. 1i, j). Consistent with the NachBac channel activity, targeted neurons showed a lower threshold for action potential firing (Supplementary Fig. 1i, k) and a higher probability for action potential firing (Supplementary Fig. 1f), compared to controls. Of note, NachBac expression also caused a lower REM in POMC neurons (Supplementary Fig. 1d), which presumably reflects a compensatory action preventing over-excitation from NachBac expression. Consistent with elevated activity of POMC neurons by NachBac expression, we also observed increased expression levels of α-MSH (Fig. 2d, f) and β-endorphin (Fig. 2e, g), compared to the contralateral un-injected side. These results, taken together, confirm that NachBac expression causes chronic activation of POMC neurons.

To examine the impact of chronic activation of POMC neurons on body weight, we next bilaterally delivered the NachBac or control AAV-

Flex-GFP virus to the Arc of POMC-Cre mice (Fig. 2h and Supplementary Fig. 3a) and monitored their body weight and feeding for 8 weeks following injections. We found no body weight differences between the NachBac and GFP groups when the mice were fed with a chow diet (Fig. 2i). To further examine whether chronic activation of POMC neurons could reverse any aspect of obesity, we first fed a cohort of POMC-Cre mice with high-fat diet (HFD) for 10 weeks (Fig. 2j), which induced obesity (Fig. 2k). These mice then received bilateral NachBac viral injections to the Arc, and we followed body weight for an additional 8 weeks. NachBac expression failed to reverse obesity but rather caused a slight, although not significant, weight gain 4 weeks after viral delivery (Fig. 2k). The additional weight gain was associated with slightly increased fat (Fig. 2l) and lean mass (Fig. 2m). We did not detect any significant changes in feeding (Fig. 2n and Supplementary Fig. 3b), $O_2$ consumption (Supplementary Fig. 3c, d), or locomotion (Fig. 2o and Supplementary Fig. 3e), which may be due to differences that are beyond our detection limits. Thus, the lack of body weight reduction in both chow and HFD-fed conditions suggests that activation of POMC neuron activity fails to reduce body weight or reverse obesity.

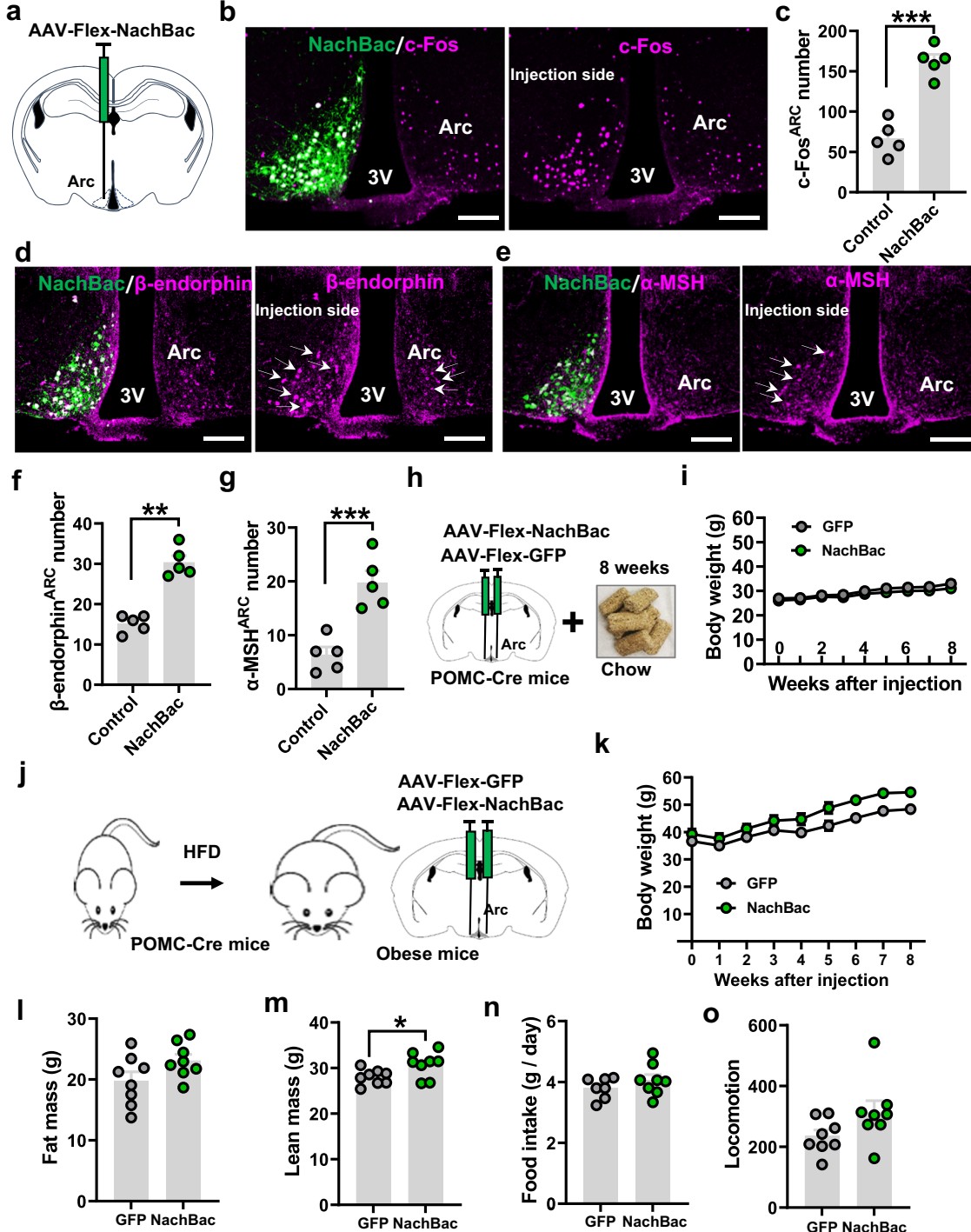

**Fig. 2 | Chronic activation of Arc POMC neurons failed to reduce body weight or obesity reversal. a** Diagram depicting one side injection of AAV-Flex-NachBac-GFP virus to Arc POMC neurons. **b** Expression pattern of viral vector (GFP) and c-Fos (left panel) or c-Fos alone (right panel) showing c-Fos induction by NachBac expression. **c** Statistical comparison of c-Fos induction between the NachBac-injected side and the non-injection side of POMC-Cre male mice ($n = 5$/each, two-tailed paired $t$-tests, ***$p = 0.0003$). **d, e** Expression patterns showing that levels of both β-endorphin (**d**) and α-MSH (**e**) were increased in the NachBac-injected Arc side of POMC-Cre male mice (arrows). Arc: arcuate nucleus, 3 V: third ventricle. Scale bar: 50 μm. **f, g** Statistical comparisons for β-endorphin (**f**, $n = 5$/each, two-tailed paired $t$-tests, **$p = 0.0011$) and α-MSH (**g**, $n = 5$/each, two-tailed paired $t$-tests, ***$p = 0.0001$) expressions. **h, i**. Diagram depicting bilateral injection of

NachBac or control GFP virus to the Arc of male POMC-Cre mice (**h**) and weekly body weight of these mice on chow feeding for 8 weeks (**i**). **j** Diagram depicting POMC-Cre mice were firstly fed HFD to induce obesity and then received NachBac injection to the bilateral Arc for body weight measurements. **k** Weekly body weight after viral delivery. **l, m** Comparison in fat mass (**l**, $n = 8$/each, two-tailed unpaired $t$-tests, $p = 0.0807$) and lean mass (**m**, $n = 8$/each, two-tailed unpaired $t$-tests, *$p = 0.0312$) between the two groups. **n, o** Comparisons between the two groups in feeding (**n**, $n = 7$/GFP and $n = 8$/NachBac, two-tailed unpaired $t$-tests, $p = 0.2791$) and locomotion (**o**, $n = 8$/each, two-tailed unpaired $t$-tests, $p = 0.0856$) measured between 2–3 weeks after viral injection when the body weight difference between the groups was minimal. All data were presented as mean ± SEM. Source data are provided in the Source Data file.

## Overexpression of α-MSH in POMC neurons fails to reduce body weight

Given the known heterogeneity of POMC neurons[45–47], no changes in body weight by chronic activation of POMC neurons may be due to non-melanocortin actions. To address this possibility, we next aimed to further examine the impact of POMC melanocortin gain of function. Since an increased α-MSH action has been considered as an increased melanocortin action[10], we generated a mouse model with selective overexpression in POMC neurons of α-MSH, which is one of the major POMC-derived peptides and considered an endogenous agonist of the melanocortin receptors known to be critical for body weight regulation[48]. Toward this, we generated a conditional AAV-Flex-α-MSH vector, in which the cDNA sequence encoding α-MSH along with the preceding signaling peptide and pro-hormone convertases (PC) cleavage site was included[49]. When we delivered this virus to one side of the Arc (Fig. 3a), we found an obvious increase of α-MSH expression in the injected side, compared to the non-injection side (Fig. 3b), whereas there were no differences in β-endorphin expression in either side (Fig. 3c, d). Thus, the α-MSH viral targeting achieves selective overexpression of α-MSH. To examine the functional consequence of the viral-mediated overexpression, we generated a cohort of POMC-Cre mice with bilateral delivery of either α-MSH virus or control GFP virus (Supplementary Fig. 4a, j) and then monitored their food intake after overnight fasting. Compared to the control group, α-MSH injected mice exhibited a significant reduction in fasting-refeeding (Fig. 3e), suggesting that α-MSH overexpression exerts an expected inhibitory action on feeding.

To examine the effect of POMC neuron-specific α-MSH overexpression on body weight gain, we delivered the virus to bilateral Arc of POMC-Cre mice followed by monitoring chow feeding for 6 weeks and HFD feeding for another 7 weeks (Fig. 3f). Compared to the control group, mice with α-MSH overexpression exhibited no significant differences in body weight fed either chow or HFD, although there was a slight reduction in body weight on HFD (Fig. 3g). In line with body weight results, we observed no differences in fat (Fig. 3h), lean mass (Fig. 3i), or food intake on either chow or HFD (Fig. 3j and Supplementary Fig. 4b). We also subjected these mice to Comprehensive Laboratory Animal Monitoring Systems (CLAMS) analysis during the transition from chow to HFD diet when there were no body weight differences between groups (Supplementary Fig. 4e), and we found no significant differences in either $O_2$ consumption (Fig. 3k and Supplementary Fig. 4c) or locomotion (Fig. 3l and Supplementary Fig. 4d) during chow or HFD feeding periods. To examine the potential effects on obesity reversal by α-MSH, we first fed POMC-Cre mice HFD to induce obesity (Fig. 3m). We then injected the virus into bilateral Arc of obese POMC-Cre mice (average body weights 50 g, Supplementary Fig. 4i), and again α-MSH mice failed to display any significant body weight changes, compared to control GFP-injected animals (Fig. 3n). Associated with no difference in body weight (Supplementary Fig. 4i), there were no alterations in fat (Fig. 3o), lean mass (Fig. 3p), feeding (Fig. 3q and Supplementary Fig. 4f), $O_2$ consumption (Fig. 3r and Supplementary Fig. 4g), or locomotion (Fig. 3s and Supplementary Fig. 4h). As the melanocortin pathway has been suggested to mediate leptin action on body weight control[50], we further investigated the role of α-MSH overexpression in reversing obesity induced by leptin deficiency. To address this, we bilaterally delivered AAV-Flex-α-MSH viruses to the Arc of adult POMC-Cre::*ob/ob* mice (Fig. 3t) and followed by monitoring their body weight for 10 weeks. Similar to diet-induced obese mice, we observed no significant impact on obesity in *ob/ob* mice with α-MSH overexpression in Arc POMC neurons, compared to controls (Fig. 3u). Consistent with this, α-MSH models showed no changes in fat (Fig. 3v) or lean mass (Fig. 3w). Collectively, these data support that α-MSH overexpression fails to reduce normal body weight gain during aging or reverse obesity induced by either HFD or leptin deficiency.

## Overexpression of β-endorphin fails to alter body weight

In addition to α-MSH as the primary endogenous MC4R agonist, β-endorphin represents the other well-studied cleaved peptide species from the POMC precursor species that affect feeding and body weight regulation[48]. Despite strong evidence suggesting a role for β-endorphin release from POMC neurons in promoting feeding, data from β-endorphin knockout mice support that its release from POMC neurons inhibits feeding, which is independent of α-MSH's feeding inhibitory action[51]. To examine the possibility that β-endorphin release from POMC neurons mediates body weight-reducing effects, we delivered AAV-Flex-β-endorphin to the Arc of POMC-Cre mice (Fig. 4a). This conditional vector was constructed by replacing α-MSH coding sequence with the β-endorphin coding sequence in the AAV-Flex-α-MSH vector. Immunostaining confirmed that the side that received the virus injection showed an obvious increase in β-endorphin expression compared to the control, non-injection side (Fig. 4b). However, there was no difference between the two sides in α-MSH expression (Fig. 4c and Supplementary Fig. 5b), suggesting the targeted and selective overexpression of β-endorphin. Given that β-endorphin release from POMC neurons has been previously shown to play a role in analgesia[52], we first tested analgesia responses and found that the β-endorphin mice exhibited a significantly less sensitive responses to pain induced via a hot plate (Fig. 4d, the latency for the mice to lick their paws), confirming that virally mediated overexpression of β-endorphin is functional.

To examine the consequence of selective β-endorphin overexpression in POMC neurons on body weight, we delivered the virus to bilateral Arc of POMC-Cre mice and monitored their body weight for 6 weeks on normal chow and a subsequent 6-week period on HFD (Fig. 4e). From mice with confirmed bilateral β-endorphin overexpression in POMC neurons (Supplementary Fig. 5a, c), we observed no significant differences in body weight between the groups when fed either chow or HFD, although we observed a trend of increased body weight on HFD (Fig. 4f). Consistent with the body weight data, there were no differences in fat mass (Fig. 4g), lean mass (Fig. 4h), or food intake (Fig. 4i). We also measured $O_2$ consumption during the chow to HFD transition with no body weight differences observed between the 2 groups (Supplementary Fig. 5d) and found no differences in feeding (Supplementary Fig. 5e), $O_2$ consumption (Fig. 4j, Supplementary Fig. 5f), or locomotion (Fig. 4k and Supplementary Fig. 5g) when mice were fed normal chow or HFD. Thus, similar to the results of both α-MSH and POMC, β-endorphin overexpression in POMC neurons also failed to reduce body weight.

## POMC overexpression does not facilitate weight reduction

In addition to α-MSH and β-endorphin, the POMC gene also gives rise to a number of additional peptides, which may also play a role in mediating body weight regulation[48]. To test this possibility, we aimed to generate mice with overexpression of POMC, having all the POMC-derived peptides elevated in POMC neurons. Toward this, we delivered conditional AAV-EF1a-Flex-mPOMC-P2A-EGFP vectors to the Arc of POMC-Cre mice. As previously observed for the other constructs, compared to the control non-injection side, viral injections showed abundant expression of GFP (Fig. 5a, b, left panels), and increased expression of α-MSH (Fig. 5a, left and middle panels) and β-endorphin (Fig. 5b, left and middle panels). Of note, the ipsilateral side of the PVH showed an increased expression of both α-MSH (Fig. 5a, right panel and Supplementary Fig. 6a) and β-endorphin (Fig. 5b, right panel and Supplementary Fig. 6b), suggesting appropriate processing and trafficking of the overexpressed peptides.

To examine the impact of POMC overexpression on body weight, we injected the POMC virus into bilateral Arc of POMC-Cre mice (Supplementary Fig. 6c). We then followed feeding and body weight for 6 weeks on normal chow and then HFD feeding for additional 6 weeks (Fig. 5c). Surprisingly, we did not observe any

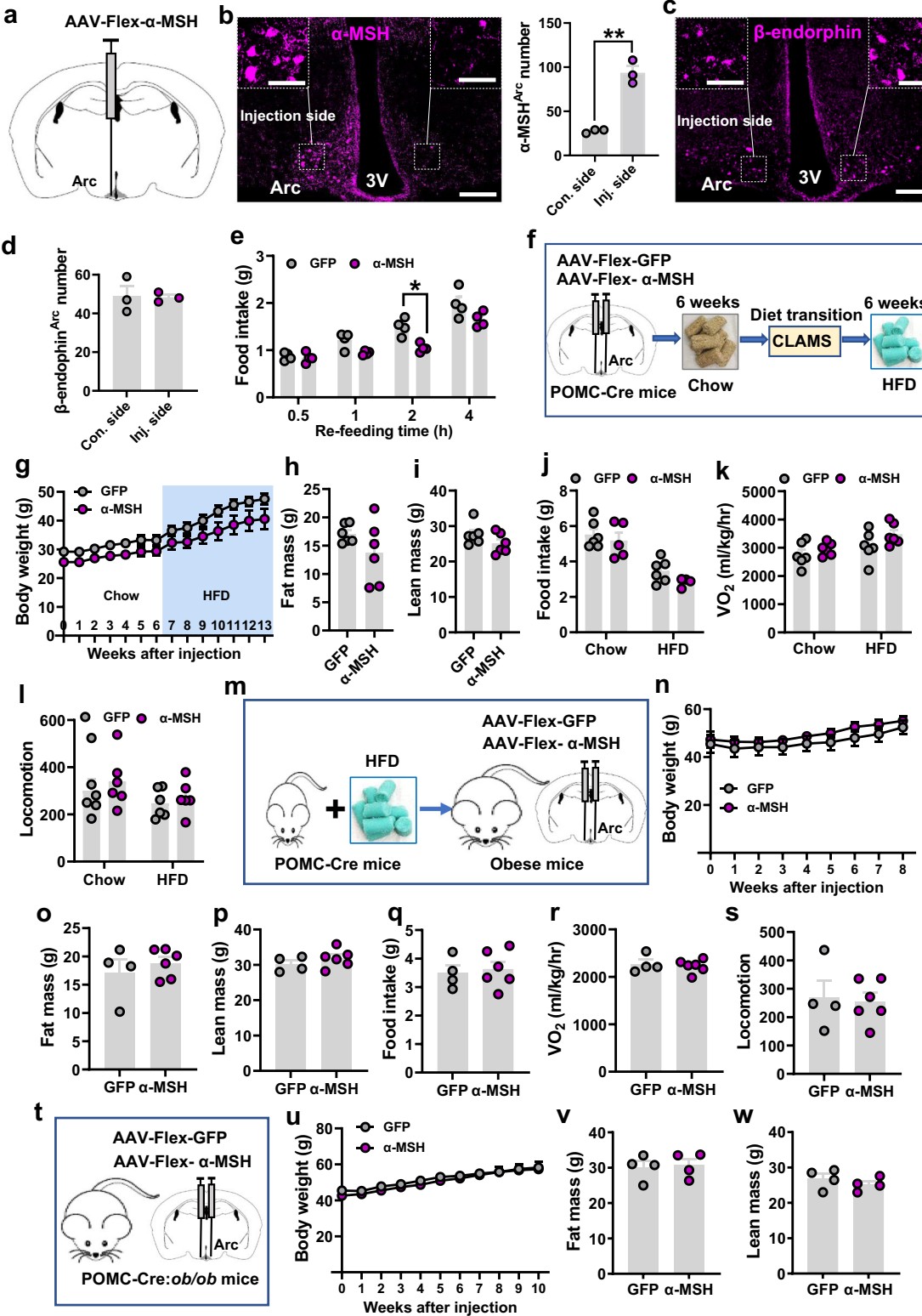

significant differences in weekly body weight of POMC-overexpressing mice, compared to control mice under either chow or HFD feeding conditions (Fig. 5d). In line with this, there were no differences in fat mass (Fig. 5e) or lean mass (Fig. 5f). We next subjected these models to CLAMS measurements during the transition from chow to HFD diet (Supplementary Fig. 6d), and we confirmed that there were no differences in food intake (Fig. 5g and Supplementary Fig. 6e) or $O_2$ consumption on normal chow. However, we

noted a significant increase in $O_2$ consumption on HFD (Fig. 5h and Supplementary Fig. 6f), which may explain the tendency for reduced body weight in POMC overexpression mice. Locomotion measured by beam breaks was not different between groups under either chow or HFD conditions (Fig. 5i and Supplementary Fig. 6g). Taken together, these data suggest that overexpression of POMC peptides in Arc POMC neurons fail to reduce body weight in chow or obesogenic HFD conditions.

**Fig. 3 | Overexpression of α-MSH in Arc POMC neurons failed to reduce body weight or obesity reversal. a–d** Expression and quantitative comparisons of α-MSH (**b**, **\*\***$p = 0.001$) and β-endorphin (**c**, **d**, $p = 0.9092$) between injection and non-injection sides ($n = 3$ mice/each), the boxed area showing the magnified expression pattern. 3 V: third ventricle, Arc arcuate nucleus. Scale bars: 50 μm for **b** and **c**, and 20 μm for boxed areas. **e** Food intake of POMC-Cre mice with either α-MSH overexpression or control viral injection after overnight fasting ($n = 4$/each, **\***$p = 0.0382$ for 2 h, $p > 0.05$ for other times). **f** Diagram depicting the experimental procedures and timelines. **g–l** Weekly body weight (**g**, $p > 0.05$ for all time points) and comparisons of fat mass (**h**, $p = 0.1867$), lean mass (**i**, $p = 0.1541$), food intake (**j**, $p = 0.7172, 0.3596$, respectively), O₂ consumption (**k**, $p = 0.6669, 0.1859$, respectively) and locomotion (**l**, $p = 0.736, 0.8697$, respectively) between the two

groups ($n = 6$/each). **m** Diagram depicting the experimental procedure for examining the obesity reversal with α-MSH overexpression in POMC neurons. **n–s** Weekly body weight (**n**, $p > 0.05$ for all time points) and comparisons of fat mass (**o**, $p = 0.4855$), lean mass (**p**, $p = 0.3275$), food intake (**q**, $p = 0.7409$), O₂ consumption (**r**, $p = 0.6872$), and locomotion (**s**, $p = 0.8297$) between the two groups ($n = 4$/GFP and $n = 6$/α-MSH). **t** Diagram depicting the experimental procedures to examine obesity reversal in *ob/ob* mice by overexpressing α-MSH in POMC neurons. **u–w** Weekly body weight (**u**, $p > 0.05$ for all time points) and comparisons of fat (**v**, $p = 0.8006$) and lean mass (**w**, $p = 0.3839$) between the two groups ($n = 4$/each). All data were presented as mean ± SEM. Two-tailed paired *t*-tests for **b** and **d**; two-way ANOVA for **e**, **g**, **k**, **l**, **n**, **u**; two-tailed unpaired *t*-tests for **h**, **i**, **p**, **q**, **r**, **s**, **v**, **w**. Source data are provided in the Source Data file.

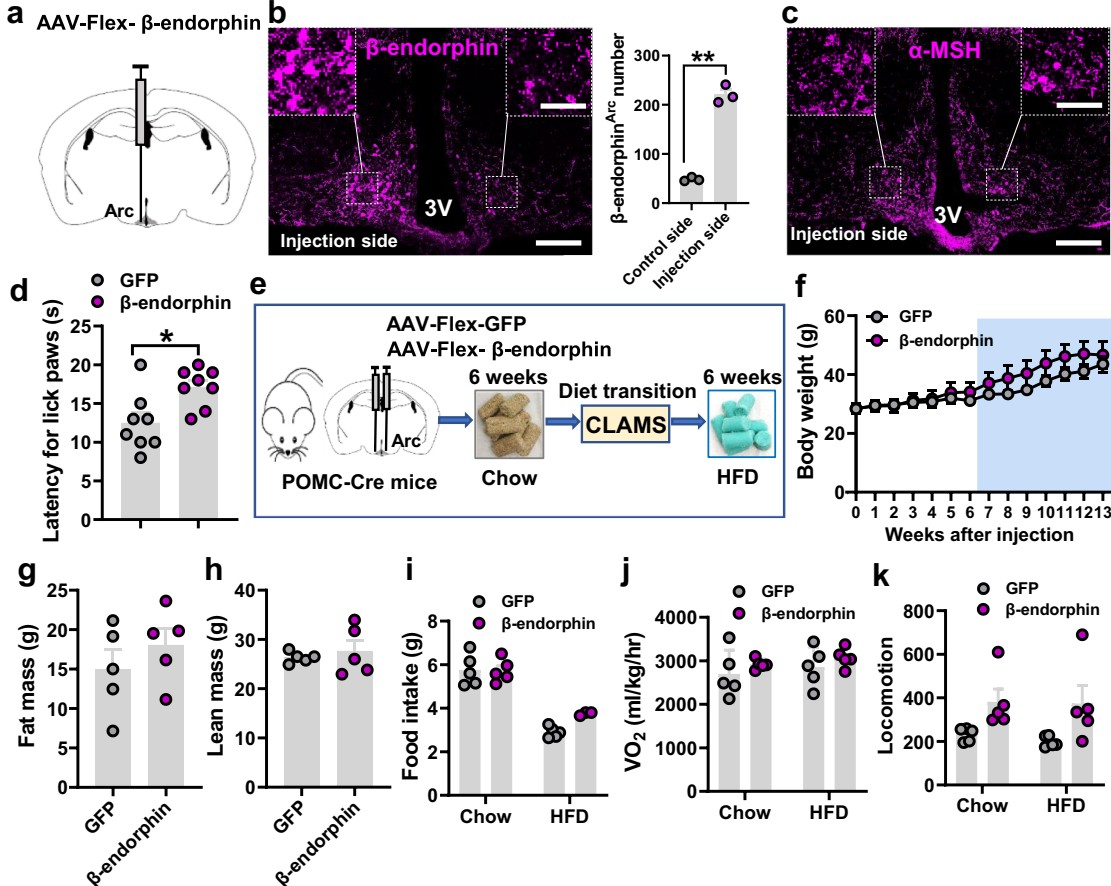

**Fig. 4 | Overexpression of β-endorphin in Arc POMC neurons failed to reduce body weight or obesity reversal. a** Diagram depicting one side injection of β-endorphin viral vectors to Arc POMC neurons. **b, c** Expression patterns of β-endorphin (**b**, $n = 3$/each, two-tailed paired *t*-tests, **\***$p = 0.0055$) and α-MSH (**c**) in the Arc of virus-injected POMC-Cre mice. Note that the β-endorphin expression level was showed much higher on the injected side compared to the non-injection side (**b**), but no difference in α-MSH expression was observed between the two sides (**c**). 3 V: third ventricle, Arc arcuate nucleus. Scale bars: 50 μm for **b** and **c** and 20 μm for boxed areas. **d** β-endorphin mice exhibited significantly less sensitivity to pain induced in hot plate assay ($n = 8$/each, two-tailed unpaired *t*-tests, **\***$p = 0. 0113$). **e** Diagram showing experimental scheme on POMC-Cre mice with bilateral delivery of the indicated viral vectors to the Arc followed by a 6-week chow diet and then

another 6-week HFD feeding with the CLAMS measurement during the diet transition. **f** Weekly body weight between the injected mouse groups after viral delivery ($n = 5$/each, two-way ANOVA, $p > 0.9$ for all time points). **g, h** Comparison in fat mass (**g**, $n = 5$/each, two-tailed unpaired *t*-tests, $p = 0.3712$) and lean mass (**h**, $n = 5$/each, two-tailed unpaired *t*-tests, $p = 0.5623$) at 13 weeks after viral delivery. **i–k** Comparison between the two groups of mice on both chow and HFD in feeding (**i**, $n = 5$/each, two-way ANOVA, $p = 0.99, 0.07$, respectively), O₂ consumption (**j**, $n = 5$/each, two-way ANOVA, $p = 0.6016, 0.6276$, respectively), and locomotion (**k**, $n = 5$/each, two-way ANOVA, $p = 0.1093, 0.0577$, respectively) measured during diet transition when there was no significant difference between the two groups. All data were presented as mean ± SEM. Source data are provided in the Source Data file.

## PVH^MC4R neurons are required but not sufficient for body weight regulation

To further verify the effect of melanocortin action on body weight regulation, we next tested the role of MC4Rs, which are known to primarily mediate melanocortin's action on body weight. As the paraventricular hypothalamus (PVH) is the major brain site that

mediates MC4R action on body weight regulation[53], we targeted PVH MC4R neurons for genetic manipulations. To directly examine the impact of manipulating the activity of PVH^MC4R neurons, we aimed to first drive chronic activation or inhibition of these neurons through the targeted expression of NachBac or Kir2.1, respectively. Toward this, we delivered AAV-Flex-NachBac-EGFP (Fig. 6a), AAV-Flex-Kir2.1-dTomato

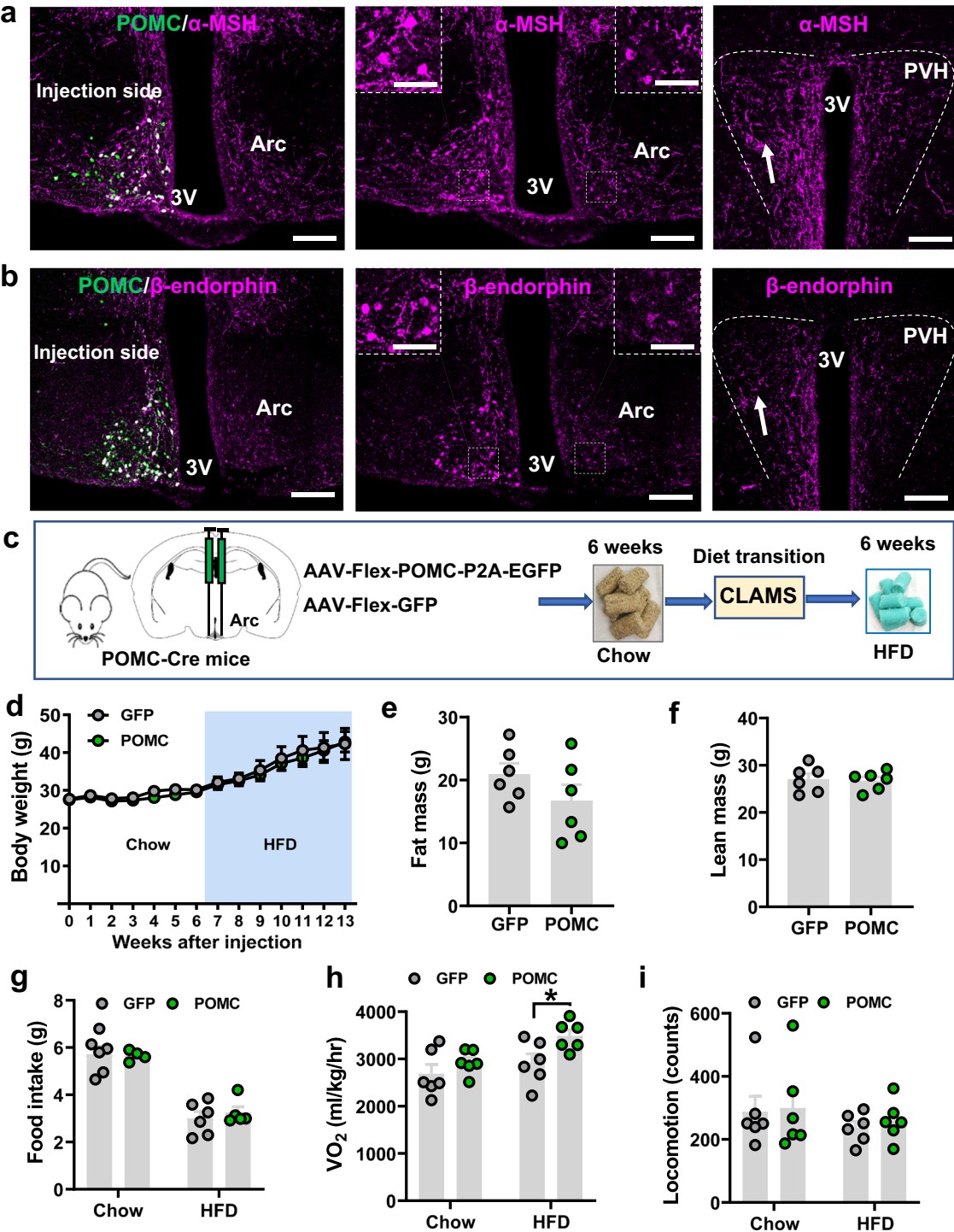

**Fig. 5 | Specific overexpression of POMC in POMC neurons failed to reduce body weight. a**, **b** POMC-Cre mice received injections of AAV-Flex-POMC-GFP viral vectors to one side of the Arc and immunostained for α-MSH (**a**, magenta) or β-endorphin (**b**, magenta). Brain sections showing colocalization between viral expression (green) and both peptides (magenta, left panels), peptide expression alone in the Arc (middle panels with a magnified view of the indicated boxed area) and the PVH (right panels). Note that the injected side exhibited an increase in both α-MSH and β-endorphin expression compared to the non-injection side. Arrows in the PVH pointing to the area with increased immunostaining structures. 3 V: third ventricle; Arc arcuate nucleus. Scale bars 50 μm for **a** and **b**, 20 μm for boxed areas. **c** Diagram showing experimental procedures and study timelines. **d** Weekly body weight after viral delivery ($n$ = 6/GFP and $n$ = 5/POMC, two-way ANOVA, $p > 0.98$ at 1–13 weeks). **e**, **f** Comparison in fat mass (**e**, $n$ = 6/each, two-tailed unpaired $t$-tests, $p = 0.1997$) and lean mass (**f**, $n$ = 6/each, two-tailed unpaired $t$-tests, $p = 0.8205$) at 13 weeks after viral delivery. **g**–**i** Comparison between the two groups of mice on both chow and HFD in feeding (**g**, two-way ANOVA, $n$ = 6/GFP and $n$ = 4/POMC on chow, $n$ = 6/GFP and $n$ = 5/POMC on HFD, $p = 0.9867$ or $0.7787$, respectively), O$_2$ consumption (**h**, $n$ = 6/each, two-way ANOVA, $p = 0.5017$, * $p = 0.0396$, respectively) and locomotion (**i**, $n$ = 6/each, two-way ANOVA, $p = 0.9749$, $0.9168$, respectively) that measured during diet transition when there was no significant difference between the two groups. All data were presented as mean ± SEM. Source data are provided in the Source Data file.

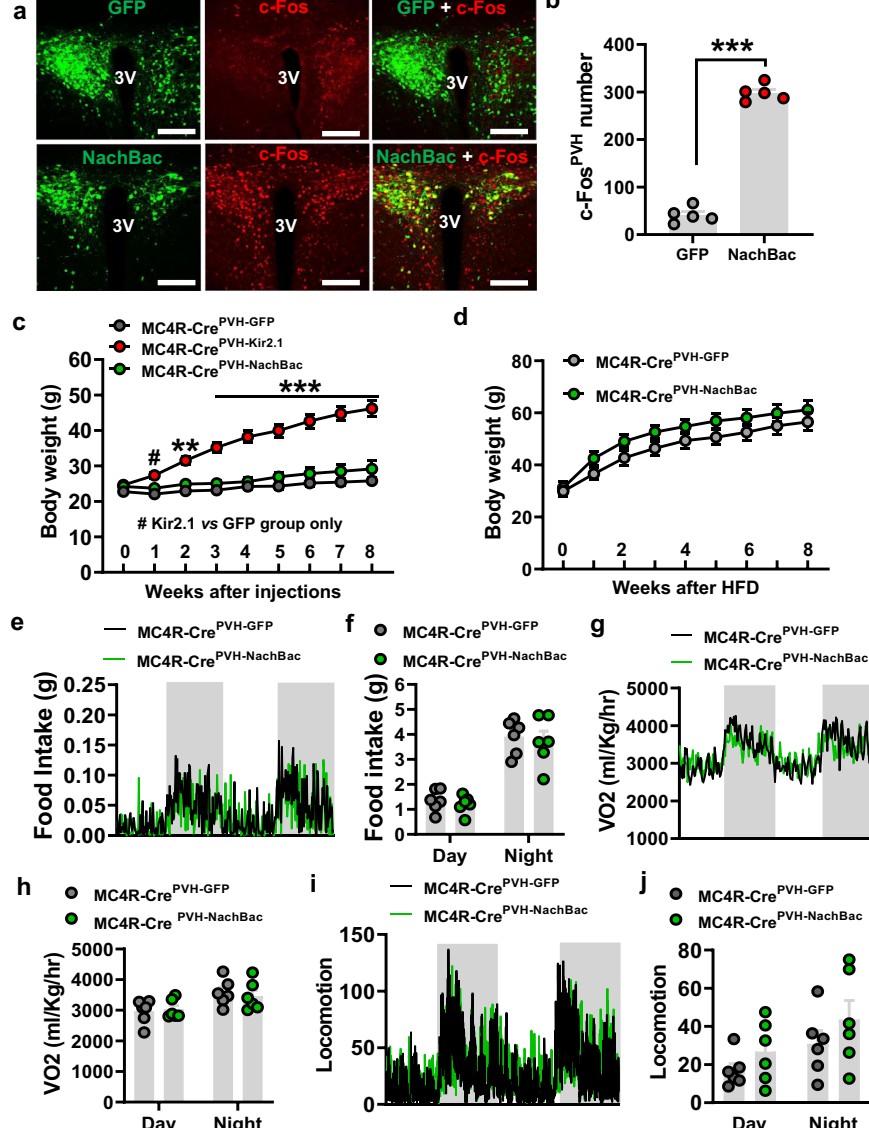

**Fig. 6 | PVH MC4Rs are required but not sufficient for body weight regulation.**
**a** MC4R-Cre mice receiving control AAV-Flex-GFP (top panels) or AAV-Flex-NachBac-GFP (bottom panels) to the PVH, and expression patterns of GFP (left panels), c-Fos (middle panels) and their merged pictures (right panels) were shown. 3 V: the third ventricle. Scale bars: 50 μm. **b** Quantification of c-Fos expression in the PVH of MC4R-Cre mice receiving injections of either control or NachBac ($n = 5$/each; two-tailed unpaired $t$-test, ***$p < 0.001$). **c** Comparison in body weight of MC4R-Cre mice receiving injections of control, AAV-Flex-NachBac-eGFP, or AAV-Flex-Kir2.1-dTomato to bilateral PVH and fed with chow ($n = 6$/GFP, $n = 5$/NachBac, and $n = 10$/Kir2.1, two-way ANOVA, #$p = 0.0292$, PVH$^{Kir2.1}$ group vs. PVH$^{GFP}$ group at 1 week; **$p < 0.01$, ***$p < 0.001$, PVH$^{Kir2.1}$ group vs. PVH$^{GFP}$ and PVH$^{NachBac}$ groups at 2–8 weeks). **d** Body weight curve of MC4R-Cre mice receiving injections of control,

AAV-Flex-NachBac-eGFP to bilateral PVH and fed with HFD ($n = 6$/GFP, $n = 5$/NachBac, two-way ANOVA, $p > 0.05$ for all time points). **e**–**j** MC4R-Cre male mice receiving injections of AAV-Flex-NachBac-GFP or control vectors to bilateral PVH were subject to CLAMS measurements 3–4 days after viral delivery when there was no difference in body weight, and shown were food intake patterns (**e**), comparisons in food intake during the day and night periods (**f**, $n = 6$/each, two-way ANOVA, $p = 0.4156, 0.4462$, respectively), $O_2$ consumption patterns (**g**) and comparisons in $O_2$ consumption during the day and night periods (**h**, $n = 6$/each, two-way ANOVA, $p = 0.9489, 0.8559$, respectively), locomotion patterns (**i**) and comparisons in bean breaks during day and night periods (**j**, $n = 6$/each, two-way ANOVA, $p = 0.5644, 0.3941$, respectively). All data were presented as mean ± SEM. Source data are provided in the Source Data file.

(Supplementary Fig. 7a) or GFP control vectors (Fig. 6a and Supplementary Fig. 7a) to the PVH of MC4R-Cre mice. Compared to controls, NachBac-injected animals exhibited abundant c-Fos expression in the PVH (Fig. 6a, b), suggesting robust activation of PVH$^{MC4R}$ neurons. To evaluate the effects of activity reduction through targeted Kir2.1 expression, we leveraged fast-refeeding analysis. Whereas control mice were induced with abundant c-Fos expression in the PVH by fast-refeeding, the same treatment induced much less c-Fos expression in Kir2.1 viral injected mice (Supplementary Fig. 7a, b), suggesting an effective action of Kir2.1 in inhibiting neuron activity. We next evaluated the physiological and feeding effects in these chronically

activated or inhibited models. Compared to controls, the Kir2.1-injected mice showed dramatically increased body weights, and by the 8th week after viral injection, they showed up to 20 g of weight gain (Fig. 6c). When measured at a time before body weight divergence, the Kir2.1-injected mice showed a clear increase in feeding (Supplementary Fig. 7c, d), reduced $O_2$ consumption (Supplementary Fig. 7e, f), and exhibited a trend in reduced locomotion (Supplementary Fig. 7g, h). In stark contrast, the NachBac-injected mice exhibited no significant difference in body weight on normal chow or HFD (Fig. 6c, d), and no alterations in feeding (Fig. 6e, f), energy expenditure (Fig. 6g, h), or locomotion (Fig. 6i, j). Together, these results suggest that

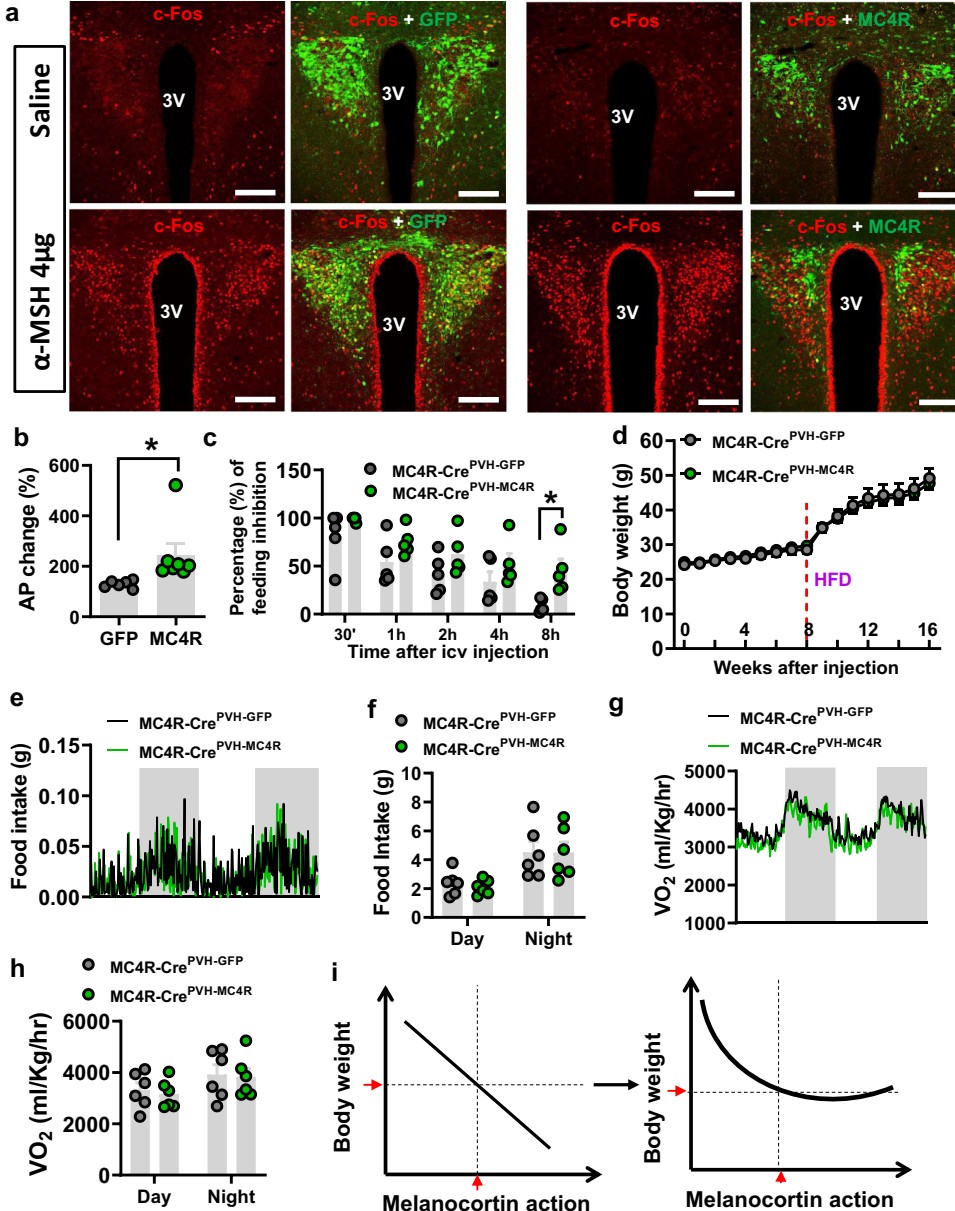

**Fig. 7 | MC4R overexpression in PVH^MC4R neurons failed to reduce body weight.**
MC4R- **a** Representative images showing AAV viral injections and associated c-Fos expression in the PVH: control vector (left panels, green), MC4R vector (right panels, green), c-Fos by i.c.v. saline (top panels, red), and α-MSH (bottom panels, red) treatments. 3 V: the third ventricle. Scale bars: 50 μm. **b** Comparison of action potential frequency changes by the MC4R agonist MTII ($n = 6$ cells/GFP from 2 mice and $n = 7$ cells/MC4R from 2 mice, two-tailed unpaired $t$-test, *$p = 0.0455$).
**c** Comparison of i.c.v. α-MSH inhibition on fast-refeeding between control and PVH MC4R overexpression mice ($n = 5$/group, two-way ANOVA, *$p = 0.0329$, PVH^GFP vs. PVH^MC4R at 8 h). **d** Weekly body weight changes of MC4R-Cre mice receiving injections of AAV-Flex-MC4R-p2A-EGFP or control vectors on chow (8 weeks) and followed by 8 weeks on HFD ($n = 4$/GFP and $n = 5$/MC4R, two-way ANOVA, $p > 0.05$ for all time points). **e**–**h** Comparisons of food intake during the 2-day measurement period (**e**) and amounts in food intake during the day and night periods (**f**, $n = 6$/each, two-way ANOVA, $p = 0.9264$, 0.9996, respectively), and O$_2$ consumption patterns (**g**) and the amounts in O$_2$ consumption during the day and night periods (**h**, $n = 6$/each, two-way ANOVA, $p = 0.9216$, 0.9653, respectively). **i** The diagram showing the proposed revision of the relationship between the strength of the melanocortin action and body weight. The perceived reverse linear relationship (left) between the melanocortin action and body weight is proposed to be replaced by a rectified relationship (right) in which lower than normal action of the melanocortin action is efficient to produce obesity, but higher than normal action is unable to efficiently reduce body weight or prevent obesity development. The red arrows on the X and Y axes indicate the respective melanocortin action and body weight at the normal basal state. All data were presented as mean ± SEM. Source data are provided in the Source Data file.

reduced activity levels of PVH^MC4R neurons lead to massive obesity, but increased activity levels of them do not alter body weight.

## MC4R overexpression in PVH^MC4R neurons fails to prevent obesity development

To examine specifically the effect of MC4R on body weight, we aimed to overexpress MC4Rs in PVH^MC4R neurons. Toward this, we injected

conditional AAV-Flex-MC4R-P2A-GFP to the PVH of MC4R-Cre mice (Fig. 7a). As expected, α-MSH, the endogenous MC4R agonist, induced significantly more c-Fos expression in the PVH of control GFP vector mice, compared to the saline retreatment group (Fig. 7a, left panel and Supplementary Fig. 8a). The same α-MSH treatment induced more c-Fos expression in the PVH of MC4R injected mice, compared to control GFP mice (Fig. 7a, right panels and Supplementary Fig. 8a). In

addition, electrophysiological recording experiments from neurons in brain slices showed that α-MSH induced more firing of MC4R-overexpressed neurons, compared to controls (Fig. 7b). As PVH MC4Rs are known to acutely inhibit feeding, we i.c.v. delivered α-MSH to examine the potential inhibitory feeding effect of MC4R activation. Overnight fasted MC4R-Cre mice with the delivery of the MC4R virus or control vectors to bilateral PVH 4 weeks prior were tested for i.c.v. saline or α-MSH. Compared to controls, α-MSH caused a significantly more inhibitory effect on fast-refeeding in the MC4R mice (Fig. 7c). Taken together, these observations confirm that overexpression of MC4Rs in the PVH causes neuron activation and acutely reduces feeding. To examine the effect of overexpressing MC4Rs in PVH[MC4R] neurons on body weight regulation, we delivered the MC4R virus to bilaterally PVH and followed weekly body weight measurements. No body weight differences were observed between the groups when fed either chow or HFD diet (Fig. 7d), which was associated with no alterations in feeding (Fig. 7e, f) or energy expenditure (Fig. 7g, h). Thus, MC4R neurons in the forms of increased neuron activity or overexpression of MC4Rs both failed to reduce body weight on chow or prevent obesity on HFD.

It is possible that no obvious impact on body weight in mice with chronic activation of POMC neurons is due to compensatory changes in downstream PVH[MC4R] neurons. Similarly, no obesity development in mice with MC4R overexpression in PVH[MC4R] neurons may be due to compensatory changes in upstream POMC neurons. To address these possibilities, we generated mice with concurrent chronic activation of POMC neurons and overexpression of MC4R in PVH[MC4R] neurons. Toward this, we delivered AAV-Flex-NachBac-GFP to the Arc and AAV-Flex-MC4R-P2A-GFP or control AAV-Flex-GFP to the PVH of 7-8 weeks old POMC-Cre::MC4R-Cre male mice (Supplementary Fig. 8b). Expression of both NachBac and MC4R viruses were confirmed in the Arc and PVH, respectively, in both groups (Supplementary Fig. 8c–e). Weekly body weight was monitored for the first 8 weeks on normal chow and followed by 8 weeks on HFD after viral delivery, and no difference in body weight was observed between the two groups (Supplementary Fig. 8f). We also performed CLAMS measurements and didn't observe an obvious difference in feeding, O$_2$ consumption or locomotion between the groups (Supplementary Fig. 8g–j). These data rule out the possibility that no impact on body weight by chronic activation of POMC neurons or by MC4R overexpression in PVH[MC4R] neurons is due to functional compensations within the melanocortin pathway.

## Discussion

Here we presented data from several complementary animal models that argue for a limited role for the melanocortin pathway in reducing body weight or obesity reversal. While chronic inhibition of Arc POMC or PVH MC4R neurons led to massive obesity, animal models with chronic activation of either group, or with specific overexpression of POMC and its derived peptides α-MSH and β-endorphin in Arc POMC neurons, or specific overexpressing MC4Rs in PVH MC4R neurons, failed to cause a significant reduction in body weight or obesity development. Given the known importance of α-MSH as the endogenous agonist of MC4Rs, we specifically examined the potential effects of selective overexpression of α-MSH in Arc POMC neurons on obesity induced by leptin deficiency or HFD, the latter mimicking the obesity condition in most human patients with obesity. Little or mild effects on obesity reversal in either case strongly argue against a significant role for an increased α-MSH in facilitating negative energy balance. Our previous studies show that chronic activation of AgRP neurons is sufficient to promote obesity while their inhibition causes no impacts on body weight or obesity reversal[43]. Taken together, bidirectional manipulations of neurons of the melanocortin pathway (POMC, AgRP, and MC4R neurons) result in severe obesity as predicted while they fail to achieve the expected obesity prevention and reversal phenotypes. Thus, the melanocortin pathway is biased toward obesity development while having no or limited roles in obesity prevention or reversal. As depicted in Fig. 7i, our overall conclusion requires a change in view on the melanocortin pathway from the well-perceived bidirectional roles in body weight regulation to being more unidirectional in driving obesity development.

Our results that chronic activation in both POMC and MC4R neurons does not affect body weight reduction or obesity prevention are somewhat surprising. It is clear that acute activation of the melanocortin pathway via either direct neuron activation or pharmacological administration of melanocortin agonists inhibits feeding, and in some cases, leads to, albeit transient and mild, body weight loss[21–24]. The discrepancy between acute and chronic manipulations may stem from redundant pathways that promote positive balance. In the Arc, GABAergic neurons, including AgRP neurons, promote positive balance in a redundant manner[43]. Acute inhibition of feeding resulting from altering one pathway can be compensated by adaptive changes in other feeding-promoting pathways over time. Adaptive compensation in energy expenditure has recently been shown to exist in humans[54]. In mice, neonatal lesion of AgRP neurons causes no gross phenotypes, while acute AgRP neuron inhibition reduces feeding[26,55]. Under this context, the starvation phenotype by i.m. toxin-induced AgRP neuron lesion in adult mice appears to be surprising. Since AgRP is also expressed in the adrenal gland[56], more studies with specific AgRP neuron lesion in the brain is warranted to specifically address this issue. Another reason for the discrepancy is that most studies on feeding inhibition use fasting conditions[21–24], which are not normally experienced in laboratory research animals fed ad libitum. Thus, the effect of acute feeding inhibition may not be reflected in chronic body weight studies in mice fed ad libitum. Therefore, acute inhibition of feeding may not necessarily translate into chronic body weight changes.

In line with our observations, within the rather rich literature on the extensively studied POMC and MC4R neurons, the vast majority of studies report an obese rather than a lean phenotype[57]. Animal models with a gain of function of orexigenic AgRP neurons develop obesity, while those with loss of function have no impact on body weight[43,58–60]. In contrast, for anorexigenic POMC and MC4R neurons, animal models with loss of function develop obesity while those of gain of function have a limited impact on body weight[11–13,28,61,62]. Specifically, for POMC neurons, the effect of neuron activation on body weight is rather controversial. Whole-body overexpression of α-MSH causes a mild effect in body weight reduction, most of which may be contributed by defective development associated with a reduced body length[49]. POMC neuron-specific gain of function in p-STAT3 surprisingly causes obesity[63]. Consistently, other studies also show that POMC neurons are capable of promoting weight gain[64,65]. An increased β-endorphin release has been suggested to mediate hyperphagia caused by cannabinoid receptors in POMC neurons[66]. POMC neuron heterogeneity may be speculated to explain inconsistent roles of POMC neurons in body weight[45,46,67]. The POMC neuron-mediated melanocortin action is still widely perceived to produce negative energy balance[37]. Our current results from several animal models with a chronic gain of function of key players within the melanocortin action convincingly demonstrate an inability of the melanocortin action in causing negative energy balance and support that the melanocortin is biased toward obesity development.

Importantly, our conclusion on a biased role for the melanocortin pathway toward promoting obesity is also supported by studies on human genetics. The vast majority of mutations in the genes implicated in the melanocortin pathway, e.g., POMC, MC4R, MRAP2, and AgRP, are associated with human obesity[68]. Of note, this is not due to a study design that biases human subjects with obesity since similar results have identified a few human gene mutations associated with leanness[69]. In particular, the MC4R loss of function of mutations

represents the most monogenic cause of human obesity[14,15,42]. Although the loss of function mutations in the MC4R gene have been identified, surprisingly, they have no impact on body weight[42]. It is worth noting that, despite an initial study suggesting that two gain of function MC4R variants are associated with body weight reduction[40], a follow-up study revealed that these mutations, in fact, have minimal impacts on body weight in mice[41].

The newly demonstrated biased role of the melanocortin pathway in driving obesity development has profound implications for the understanding of normal body weight regulation. Current debates on the mechanisms underlying human predisposition to obesity development centers on the interaction between genetic differences and environmental changes with the initially proposed "thrifty gene" and recently "drifty gene" hypotheses[6,8]. Although the underlying genes in these hypotheses remain to be identified, the biased role of the melanocortin pathway, as a chief body weight regulator in the brain, may serve as a neural basis for the predisposition to obesity development. Facing the same environmental changes, those with genetic makeups that favor reduced melanocortin action will be highly susceptible to obesity development, while others with genetic makeups that favor increased melanocortin action will not be able to reduce body weight, which eventually leads to a one-directional predisposition to obesity.

The bias towards obesity via the melanocortin pathway may also underlie the observed drastic differences in the impact on physiology between low and high leptin levels. Leptin resistance has been suggested to be a major culprit for obesity development[31]. However, recent studies suggest that leptin action is actually increased in obesity, and thus argue against the role of leptin resistance as a causal role in obesity[31–33,70]. Given the known role of leptin in activating POMC neurons and inhibiting AgRP neurons, reduced leptin action accompanied by reduced POMC neuron activity and increased AgRP neuron activity is predicted to promote positive energy balance. Indeed, conditions with reduced leptin levels are known to be associated with hunger, e.g. fasting, or obesity development, e.g., leptin deficiency. In this sense, leptin essentially functions as a starvation hormone owing to much more profound effects induced by lowering leptin levels during fasting[33,71]. However, increased leptin levels, accompanied by increased POMC neuron activity and reduced AgRP neuron activity, are predicted to have a limited impact on energy balance. In line with this, leptin supplementation has failed to treat obesity[33]. Moreover, specific restoration of leptin receptors in POMC neurons (presumed activation of these neurons) or AgRP neurons (presumed inhibition of these neurons) causes little effects on obesity reversal in db/db mice[72,73]. Thus, the biased role of the melanocortin pathway may also serve as a neural basis for the observed potent impact on positive energy balance by reduced leptin levels with no obvious effect on body weight by increased leptin levels.

The biased role of the melanocortin action toward obesity development calls for alternative approaches toward viewing MC4Rs as therapeutic targets against the current obesity epidemic. Extensive efforts have been made to generate specific therapeutic drugs that activate MC4Rs towards treating obesity. These efforts have not been successful and the sustained effects from MC4R agonists on body weight have been difficult to achieve[36,74,75]. In fact, previous rodent studies suggest that transgenic whole-body overexpression of α-MSH causes only minor reductions in body weight or obesity prevention, the majority of which may be due to reduced body mass associated with reduced body length occurred during development in the transgenic animals[49]. Thus, the failure of efforts to generate an effective treatment based on the MC4R action against obesity may largely stem from the inability of MC4R action on reducing weight rather than the effect on MC4R activation. Notably, evidence from both humans and rodents supports the efficacy of precision treatment, i.e., the treatment effect is to reconstitute the obesity-causing defects[76]. In fact, the recently approved FDA anti-obesity drug Setmelanotide is designed to activate MC4Rs in selective patients with obesity known to harbor mutations in key genes involved in the melanocortin pathway[37]. Given the demonstrated biased and redundant role of neural pathways in promoting obesity[43], precision therapeutics against obesity may represent an important alternative but more effective strategy to reverse the current obesity epidemic.

## Methods

### Animals
Mice were housed under a 12 h light/dark cycle within a temperature-controlled room (21–22 °C) and allowed to free access food and water. All animal research followed relevant ethical regulations. Animal care and use procedures were approved by the Animal Welfare Committee of The University of Texas Health Science Center at Houston. POMC-Cre, MC4R-Cre, and ob/+ mice were obtained from Jax Laboratory and maintained in local colonies through breeding. All mice used for stereotaxic injections were at least 8–10 weeks old at the time of surgery.

### Stereotaxic injections and viral vectors
Stereotaxic surgeries to deliver viral constructs were performed as previously described in ref. 43. Briefly, mice were anesthetized with a ketamine/xylazine cocktail (100 and 10 mg/kg, respectively), and their heads were affixed to a stereotaxic apparatus. Viral vectors were delivered through a 0.5 μL syringe (Neuros Model 7000.5 KH, point style 3; Hamilton, Reno, NV, USA) mounted on a motorized stereotaxic injector (Quintessential Stereotaxic Injector; Stoelting, Wood Dale, IL, USA) at a rate of 40 nL/min. Viral preparations were made by the Baylor NeuroConnectivity Core with the sterotype DJ8 and titers more than $10^{12}$ vg/mL. Viral delivery was targeted to the Arc or PVH through four local injections with two to each side (200 nL/side, anteroposterior (AP): −1.4 and −1.6 mm, mediolateral (ML):±0.2 mm, and dorsoventral (DV): −5.9 mm). AAV-EF1a-Flex-EGFP-P2A-mNachBac, AAV-EF1a-Flex-Kir2.1-P2A-dTomato, AAV-CAG-Flex-mPOMC-P2A-EGFP, AAV-CAG-Flex-Alpha-MSH, or AAV-CAG-Flex-Beta-endorphin, or AAV-CAG-Flex-MC4R-P2A-GFP were delivered bilaterally or unilaterally into the Arc of POMC-Cre mice, POMC-cre:ob/ob mice, or the PVH of MC4R-cre mice. Two mutations (E224G and Y242F) were made in the Kir2.1 construct so that the channel will be more effective in reducing neuron activity[43]. AAV-Flex-GFP or AAV-Flex-mCherry injections were used as a control group.

### Brain slice electrophysiological recordings
Acute brain slices containing the arcuate nucleus or the PVH were prepared from POMC-cre mice that had received stereotaxic injections of AAV- Flex -mNachBac, AAV-Flex-Kir2.1, or AAV-Flex-GFP into the Arc, or MC4R-cre mice that received injections of AAV-Flex-MC4R or AAV-Flex-GFP into the PVH about 1–4 weeks prior to the recording. Mice were deeply anesthetized with a ketamine/xylazine cocktail (i.p) and transcardially perfused with an ice-cold cutting solution containing the following (in mM): 75 sucrose, 73 NaCl, 26 NaHCO₃, 2.5 KCl, 1.25 NaH₂PO₄, 15 glucose, 7 MgCl₂, and 0.5 CaCl₂, saturated with 95% O₂/5% CO₂. Coronal slices (280 μm) were cut in oxygenized, ice-cold cutting solution, and incubated for 30 min in artificial cerebrospinal fluid (aCSF) containing (in mM): 123 NaCl, 26 NaHCO₃, 2.5 KCl, 1.25 NaH₂PO₄, 10 glucose, 1.3 MgCl₂, 2.5 CaCl₂ bubbling with 95% O₂/5% CO₂, at 32–34 °C, then maintained at room temperature for at least 1 h to allow for recovery before any electrophysiological recordings. Individual slices were then transferred to a recording chamber mounted on an upright microscope (Olympus BX51WI) and continuously superfused (2 mL/min) with aCSF maintained at 32–34 °C by passing it through a feedback-controlled in-line heater (TC-324B; Warner Instruments). Cells were visualized through a 40X water-immersion objective with differential interference contrast (DIC) optics and infrared illumination. Fluorescent-guided whole-cell patch

clamp recordings were performed with a MultiClamp 700B amplifier (Axon Instruments). Patch pipettes were 3–5 MΩ when filled with an internal solution containing (in mM): 142 K-gluconate, 10 HEPES, 1 EGTA, 2.5 MgCl₂, 0.25 CaCl₂, 4 Mg-ATP, 0.3 Na-GTP, 10 Na₂-Phosphocreatine (pH 7.3 adjusted with KOH, 300 mOsmol). Immediately after the formation of the whole-cell configuration, membrane potential, and spontaneous firing were obtained before a negative current pulse was applied to measure the input resistance and a series of current steps were used to detect the action potential rheobase (the minimal injected current required for the generation of an action potential). To access the effect of MTII, the firing rate of PVH MC4R neurons was set to ~1 Hz at baseline. After a 2-min baseline recording, aCSF containing 100 nM MTII was bath applied until a plateau response was obtained. Firing rates were expressed as percent change relative to baseline values. Only recordings with a series resistance less than 20 MΩ were used.

### Body weight studies
Weekly body weight was monitored on all mice fed standard mouse chow (Teklad F6 Rodent Diet 8664, 4.05, 3.3 kcal/g metabolizable energy, 12.5% kcal from fat, Harlan Teklad, Madison, WI) for 10 weeks after viral delivery. For the diet transition experiment, the mice were fed a chow diet for 6 weeks and switched to a high-fat diet (HFD, Research Diets, New Brunswick, NJ, D12492) for 7 weeks. Body composition (fat mass and lean mass) was measured at indicated times by using the Echo-MRI system (Echo MRI, Houston, Texas). For obesity reversal experiments, POMC-Cre male mice were first fed with HFD for 3 months to induce obesity (more than 45 g of body weight) before viral delivery and weekly body weight was followed after viral delivery for additional 8 weeks.

### Food intake measurements
Mice were individually housed for at least 4 days before daily food intake was measured. Food intake was measured 2–4 weeks after viral delivery when there was no difference in body weight between the measurement groups. Daily food intake was calculated as the mean values of one-week measurements. For acute fasting-refeeding, mice were individually housed and acclimated for 4 days before feeding experiments. Refeeding was monitored at 0.5, 1, 2, and 4 h after overnight fasting. For the MTII feeding inhibition experiment, male mice receiving AAV-flex-MC4R-P2A-GFP or control vector injections were also implanted with a cannula to the lateral ventricle. After a 1-week recovery, these mice were used for fast-refeeding experiments with i.c.v. treatments of either saline or MTII (4 μg/2 μl) and food intake was measured for 0.5, 1, 2, 4, and 8 h.

### CLAMS analysis
Energy expenditure was measured by oxygen consumption with indirect calorimetry. Mice were placed at room temperature (22–24 °C) in chambers of a Comprehensive Lab Animal Monitoring System (CLAMS, Columbus Instruments, Columbus, OH) with the capacity of simultaneous measurement of food intake, O₂ consumption, and locomotion (beam breaks). Food and water were provided ad libitum. Mice were acclimatized in the chambers for at least 48 h prior to data collection. Readings of O₂ consumption, locomotion, and food intake were plotted and compared between groups. For those mice with obese phenotypes (POMC-Cre or MC4R-Cre mice with Kir2.1 viral injections), mice were subject to the CLAMS measurement 3–4 days after viral delivery when there was no significant difference in body weight between groups. For those mouse groups without body weight difference, CLAMS measurements were performed during the diet switch with the first 3 days on chow followed by another 3 days on HFD to achieve simultaneous measurements on both chow and HFD diets.

### Leptin treatment
For the leptin treatment experiment, mice with stable obesity 10 weeks after Kir2.1 viral delivery received the implantation of 14-day duration minipumps (DURECT Corporation, Cupertino, CA), prefilled with leptin (Dr. Parlow, Harbor-UCLA, CA), which allows a slow infusion of leptin (50 ng/h) for 13 days. Mice were measured for feeding and body weight every 3 days. *Ob/ob* mice were used as a positive control group for leptin treatment.

### Hot plate test
The hot plate test is to determine the antinociceptive effect of acute thermal nociception as previously described for the effect of β-endorphin[52]. Briefly, mice received the β-endorphin viral injections to the Arc of POMC-Cre male mice (200 nl/side) were acclimated first in the test room for 30 min and then placed in the testing chamber for 1 min each day for 3 days. On the testing day, the inner bottom surface of the testing chamber was set at 50 ± 2 °C, and mice were then placed into the chamber. The time (in seconds) that mice take to either lick or flick the hind paws, or jump off was recorded. Mice not responding within the 20 s set period were removed to avoid potential tissue damage and assigned a score of 20 s.

### Immunostaining and imaging
After behavioral and metabolic experiments were completed, study subjects were anesthetized with a ketamine/xylazine cocktail (100 and 10 mg/kg, respectively) and transcardially perfused with 0.9% normal saline followed by 10% formalin. Freshly fixed brains were then harvested and placed in 10% buffered formalin at 4 °C overnight for post-fixation, then dehydrated in 30% sucrose solution. Brains were frozen and sectioned into 30-μm thickness slices with a sliding microtome and mounted onto slides for post hoc visualization of injection sites and cannula placements. Brain sections were immunostained with the following primary antibodies: Rabbit anti-c-Fos (1:1000, #2250 S, Cell Signaling Technology, Danvers, MA), Rabbit anti-p-STAT3 (1:1000, #9145 S, Cell Signaling Technology, Danvers, MA), Rabbit anti-α-MSH (1:1000, #H-043-01, Phoenix Pharmaceuticals, CA, USA), and Rabbit anti-β-endorphin (1:1000, #H-022-33, Phoenix Pharmaceuticals, CA, USA). Then, all brain sections were incubated with AlexaFluor 488 conjugated donkey anti-rabbit IgGs (1:200, Jackson ImmunoResearch Laboratories; LOT # 110898). Brain sections with reporter expression and/or immunostained fluorescence were visualized with confocal microscopy (Leica TCS SP5; Leica Microsystems, Wetzlar, Germany). c-Fos were counted from three matched sections containing rostral, middle, and caudal of Arc or PVH from individual mice, the counted numbers were averaged and compared between conditions and/or groups.

### Statistical analysis
All data were presented as mean ± SEM. Comparisons among groups were performed using Student's *t*-test or one-way or two-way ANOVA followed by Tukey's or Šídák's multiple comparison post hoc tests by using GraphPad Prism.9 (GraphPad Software, La Jolla, CA). $P < 0.05$ was considered significant in all cases. For data where representative micrographs are shown, we repeated each experiment independently with similar results as follows: Figs. 1b = 3 animals; 1d = 8 animals; 2b, d, e = 5 animals; 3b, c = 3 animals; 4b, c = 3 animals; 5a, b = 5 animals; 6a = 5 animals; 7a = 5 animals. Supplementary Figs. 1f = 4 animals for lean, Kir2.1 + saline and Kir2.1 + leptin groups, and 3 animals for *ob/ob* + leptin group; 3a = 7 animals; 4a = 5 animals; 5a = 5 animals; 6c = 6 animals; 7a = 5 animals; 8c, d, e = 5 animals.

### Reporting summary
Further information on research design is available in the Nature Portfolio Reporting Summary linked to this article.

## Data availability

All data generated or analyzed during this study are included in this published article. Source data are provided with this paper.

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

## Acknowledgements

This study was supported by NIH R01 DK136284 and 1R01DK131446 (Q.T.), R01DK109934 and DOD W81XWH-19-1-0429 (Q.T. and B.R.A.); NIH R01 DK120858 (Q.T. and Yo.X.); R01DK117281 and R01DK101379 (Yo.X.). We also acknowledge the Neuroconnectivity Core funded by NIH IDDRC grant 1 U54 HD083092 and Baylor College of Medicine Gene Vector Core for providing AAV vectors. HL was supported by the Graduate Student Overseas Study Program from Shanghai University of Chinese Traditional Medicine. We would like to acknowledge the Tong lab members for their helpful discussion. Q.T. is the holder of the Cullen Chair in Molecular Medicine at McGovern Medical School.

## Author contributions

H.L. conducted the research with help from Yu. X., Y.J., Z.J., J.O.-G., Z.M., J.C.M., and J.C.; J.O.-G., B.R.A., and Yo. X. provided essential reagents; Q.T. and C.H. conceived and designed the experiments and wrote the manuscript with significant inputs from all authors.

## Competing interests

The authors declare no competing interests.
