## [Peer Review File · Nature Communications]

The Melanocortin Action Is Biased Toward Body Weight GainReviewers' comments:

Reviewer #1 (Remarks to the Author):

The manuscript by Li et al. addresses the impact of increasing the activity of the melanocortin pathway on energy balance. The melanocortin pathway includes POMC-expressing neurons in the arcuate nucleus (ARC) and MC4R-expressing neurons in the paraventricular hypothalamus (PVH). It is well known that mutations in the MC4R gene cause severe obesity in humans and mice and are responsible for a significant fraction of child morbid obesity. Recent optogenetic and chemogenetic studies have shown that acute inhibition of POMC neurons in the ARC or MC4R neurons in the PVH stimulates food intake, whereas acute activation of these neurons suppresses food intake. These findings lead to an exciting but untested concept that pharmacologic or genetic activation of the melanocortin pathway could be effective therapeutic interventions for obesity. In this study, the authors extensively tested this concept by chronically activating the pathway in mice through expression of the bacterial sodium channel NachBac in POMC neurons or PVH MC4R neurons, overexpression of POMC or its derivative polypeptides (alpha MSH or beta endorphin) in ARC POMC neurons, or overexpression of MC4R in the PVH MC4R neurons. Their data show that chronic inhibition of the melanocortin pathway leads to massive obesity, as expected; however, they found that chronic activation of the melanocortin pathway neither prevented obesity development nor reversed obesity. These findings make the authors conclude that “the melanocortin action drives one-directional obesity development”. In general, the experiments were well designed, viral vectors were thoroughly validated, and the data are of high quality. The following minor issues must be addressed before the manuscript is ready for publication.

1. The title, the abstract, and the main text state that the role of the melanocortin pathway is biased to obesity development. It is difficult to believe that one pathway is formed during evolution mainly for the development of a disease state. It may be more appropriate to use a phrase like protection of weight loss to describe the role of the melanocortin pathway.
2. Supplementary Figure 2: The authors state that leptin treatment “dramatically increased p-STAT3 expression in the Arc (Supplementary Fig. 2f). However, the three right images look similar. The authors should replace them with more representative images. Ideally, p-STAT2 immunoreactivity should be quantified in some ways.
3. Figure 5a and b: The difference in alpha-MSH or beta-endorphin immunoreactivity in the PVH between the ipsilateral side and contralateral side is not apparent. New images are needed.
4. It does not appear that mice gained more weight on a high-fat diet than on a chow diet (Fig. 3f, 4f) in some experiments.

Baoji Xu

Reviewer #2 (Remarks to the Author):

In this manuscript Li et al reinforce findings that inhibition of POMC or MC4R neurons of the PVN leads to overeating and obesity. They then show that unlike chronic inhibition, chronic activation fails to reduce long-term body weight. Overall these studies appear well-executed but the advancement toward our understanding of the signaling mechanisms or function of the neurons is incremental as much of the inhibitory data has been demonstrated previously using different methodology and the activation data is not particularly surprising given the need for redundant circuitry to ensure starvation is avoided. Nonetheless, there are a few places where I've made suggestions that would strengthen the overall conclusions.

I'd remove adverbs such as "unexpectedly" in the abstract or "surprisingly" in the results section as the dogma in the energy homeostasis field is that redundancy networks involved in keeping the animal eating are required to maximize survival. Thus, although we often associate POMC and MC4R neurons with satiety, it is not shocking that taking them offline does not significantly alter feeding and body weight as parallel circuits are built in to maintain energy balance.

The abstract also makes a bold statement about "the largely failed efforts in identifying effective therapeutics toward treating general obesity via activation of the melanocortin action," when this is simply not true. Setmelanotide has had phenomenal success in altering feeding and body weight. True, while it has not been granted governmental permission to be tested on patients with general obesity, there is very little evidence suggesting it won't be just as successful as observed in patients with melanocortin mutations. Furthermore, additional compounds targeting this system have been shown to have similar weight-reducing properties in pre-clinical studies.

Kir2.1-inhibition of POMC neurons recapitulates both the short and long-term hyperphagia and weight gain previously reported with POMC inhibition/ablation so this does not add much to what is known.

Unless there is a difference observed between males and females, I'd suggest combining the sex data into one graph instead of two separate ones for each measurement.

One of the biggest weaknesses of the manuscript is the lack of verification of overexpressed proteins. While immunohistochemistry is performed (but not quantified; more on this below), additional assays would be required to ensure proper functionality beyond what is shown. For example, n=4 to show that at a very specific time point (2 hours) that re-feeding is lower confirms functionality of alpha-MSH is weak.

The authors often use representative images to highlight a point of text but without proper quantification, it is hard to decipher and becomes hard to believe. For example, the alpha-MSH staining in Figure 3b on the non-injection side appears extremely low at baseline when the authors want to demonstrate that injection of the construct leads to robust expression in comparison but then baseline alpha-MSH appears high in the control injected with beta-endorphin in Figure 4c.

Similar to the POMC experiments, prior work has demonstrated that acute inhibition of MC4R PVN neurons or specific knockout of Mc4r drive obesity.

We would like to thank both reviewers for their critical reviews, which really helped improve the manuscript. Here we provide point-by-point responses to reviewers' concerns.

Reviewer 1

The manuscript by Li et al. addresses the impact of increasing the activity of the melanocortin pathway on energy balance. The melanocortin pathway includes POMC-expressing neurons in the arcuate nucleus (ARC) and MC4R-expressing neurons in the paraventricular hypothalamus (PVH). It is well known that mutations in the MC4R gene cause severe obesity in humans and mice and are responsible for a significant fraction of child morbid obesity. Recent optogenetic and chemogenetic studies have shown that acute inhibition of POMC neurons in the ARC or MC4R neurons in the PVH stimulates food intake, whereas acute activation of these neurons suppresses food intake. These findings lead to an exciting but untested concept that pharmacologic or genetic activation of the melanocortin pathway could be effective therapeutic interventions for obesity. In this study, the authors extensively tested this concept by chronically activating the pathway in mice through expression of the bacterial sodium channel NachBac in POMC neurons or PVH MC4R neurons, overexpression of POMC or its derivative polypeptides (alpha MSH or beta endorphin) in ARC POMC neurons, or overexpression of MC4R in the PVH MC4R neurons. Their data show that chronic inhibition of the melanocortin pathway leads to massive obesity, as expected; however, they found that chronic activation of the melanocortin pathway neither prevented obesity development nor reversed obesity. These findings make the authors conclude that "the melanocortin action drives one-directional obesity development". In general, the experiments were well designed, viral vectors were thoroughly validated, and the data are of high quality. The following minor issues must be addressed before the manuscript is ready for publication.

Response: We thank the reviewer's appreciation on the significance and quality of this work.

1. The title, the abstract, and the main text state that the role of the melanocortin pathway is biased to obesity development. It is difficult to believe that one pathway is formed during evolution mainly for the development of a disease state. It may be more appropriate to use a phrase like protection of weight loss to describe the role of the melanocortin pathway.

Response: We agree with the reviewer's comments. We have changed the title to: "The melanocortin Action is Biased Toward Body Weight Gain".

2. Supplementary Figure 2: The authors state that leptin treatment "dramatically increased p-STAT3 expression in the Arc (Supplementary Fig. 2f). However, the three right images look similar. The authors should replace them with more representative images. Ideally, p-STAT2 immunoreactivity should be quantified in some ways.

Response: We feel sorry for the confusion in the way we displayed our results. As instructed, we have updated pictures with detailed quantification on pSTAT3 immunostaining.

3. Figure 5a and b: The difference in alpha-MSH or beta-endorphin immunoreactivity in the PVH between the ipsilateral side and contralateral side is not apparent. New images are needed.

Response: As instructed, we have provided new pictures and also used arrows to indicate the regions with increased fibers.

4. It does not appear that mice gained more weight on a high-fat diet than on a chow diet (Fig. 3f, 4f) in some experiments.

Response: We feel sorry that due to different scales of the y axis used in the panels, the numbers looked deceptively different. The following are raw data from both panels.

Raw data of Fig. 3f

	GFP						α-MSH					
0	28.4	31.9	29.3	29.4	25.9	30.4	25.3	27.9	29.8	25.4	23.4	21.6
1	27.9	30.8	28.8	30.2	26.6	30.6	25	28.3	29.6	24.5	24.1	22
2	28.8	33	30.3	30.9	27.2	31.2	25.5	28.8	31.5	27.9	24.8	23
3	29.8	32.8	31.3	34.8	28.2	31.9	26.2	29.7	32.5	28.1	26	23.8
4	29.3	33	31.2	37.8	28.3	33.2	26.8	29.9	33.1	28.5	26.7	24.1
5	29.9	34.1	32.6	39.9	28.7	34.5	27.6	31.2	35.8	29.1	26.8	24.8
6	29.9	33.7	32.3	41.2	30.2	31.9	28.4	30.7	36.2	29.2	27.7	23.9
7	34	37.9	36.7	43.3	32	35.2	31.2	35.9	42	31.2	28.9	24.5
8	34	38.4	37.3	46.1	33.6	35.5	33.3	36	40.4	31.1	30	24.8
9	35.7	41.5	39.1	49	35.4	39	34.2	38.9	44.2	30.8	33.7	25.2
10	38.4	44.4	42.4	51.5	39.3	43.5	36.6	42.3	48.1	32.3	32	26.8
11	40.5	47.1	45	51.8	41.3	47.5	38.6	45.8	50.4	33.3	34.4	28.7
12	41.7	48.3	47.2	52.5	41.6	48.2	39.3	47.6	52.9	33.5	35.9	30.2
13	41.9	49.3	49.2	52.9	41.8	50	39.4	46.9	54.7	35.1	36.7	30.6

Raw data of Fig. 4f

	GFP					β-endorphin				
0	28.4	27.1	28.3	30.4	27.5	25.1	23.6	28.7	32.2	32.6
1	30.2	28.3	30.1	30.6	28.2	25.8	24.3	30	34.9	31.5
2	29.7	28.9	30.3	31.2	28.6	25.4	23.8	29.5	34.9	33.4
3	30.7	30.2	30.8	31.9	28.8	26.7	25.1	31.2	37.5	35.5
4	29.6	29.9	31.2	33.2	29.6	26.2	24.7	30.9	38.9	38
5	30.2	31.4	32.8	34.5	30.7	27.2	25.2	33.1	43.5	40.2
6	31	30.7	31.4	31.9	30.6	28.2	27.7	32.1	43.7	38.6
7	32	33.2	35.5	35.2	30.6	27.9	31.5	38.2	48.1	39.7
8	33.1	31.8	35.5	35.5	31.1	28.5	31.3	38.2	51.7	43.9
9	33.4	32.1	36.2	39	33.2	29.4	35	37.3	53.8	46.5
10	34	34.8	39.7	43.5	36.8	31.8	39.6	40.8	55.4	51.7
11	34.8	35.9	42.6	47.5	40.1	33.7	41.8	44.4	56.5	54.2
12	35.4	36.1	45.1	48.2	40.9	34.6	42.9	45.6	57.4	54.5
13	35.9	39.6	48.6	50	43.5	33.7	39	48.1	58.2	54.2

We hope that the review agrees that these numbers on chow and HFD are both within in a similar range.

Reviewer #2 (Remarks to the Author):

1. I'd remove adverbs such as "unexpectedly" in the abstract or "surprisingly" in the results section as the dogma in the energy homeostasis field is that redundancy networks involved in keeping the animal eating are required to maximize survival. Thus, although we often associate POMC and MC4R neurons with satiety, it is not shocking that taking them offline does not significantly alter feeding and body weight as parallel circuits are built in to maintain energy balance.

Response: We thank the reviewer for raising this important point on the significance. Although it has been conceived to some degree in the field on the concept of "as the dogma in the energy homeostasis field is that redundancy networks involved in keeping the animal eating are required to maximize survival", we feel it is unfair to use this seemingly philosophical statement to assess the significance of this study. Does this mean that, since the hypothalamus is believed to regulate

feeding in general, any studies in the hypothalamus on feeding will be predicted and not significant? In fact, there has been few effective studies to test this redundant theory. Importantly, **this is absolutely not the case for the function of the melanocortin action (see details below)**. Thus, we have to respectfully disagree with this reviewer on the negative assessment on our data on activation and gain-of function of the melanocortin action with no obvious effects on reducing body weight. We use the following 5 points to address this issue.

- a) **Significance of the predicted data on inhibition of POMC and MC4R neurons**. As our title suggests, the main argument of this study is to show that the melanocortin action is biased toward weight gain. In order to validate this claim, we hope that the reviewer would agree that it is essential to demonstrate “bias”, i.e. inhibition causing obesity while activation causing little effects. In other words, in an analogy to a controlled experiment, the purpose of showing the data from the inhibition experiments is to demonstrate a positive control effect, without which, the significance on the experimental group, i.e. the activation data, will not stand. Importantly, the inhibition data were generated using a totally different approach from those used previously, and therefore also serve to validate that the approaches we used in the current study are robust. Thus, in this study, the purpose of the inhibition data is not to demonstrate the novelty of the data, but instead to contrast the effect of negative data from the activation or gain of function approaches. **The major trust of this study is the activation and gain of function of the melanocortin action fails to reduce body weight and reverse obesity**. Due to these reasons, we feel it is important to simultaneously show both sets of data with contrasting effects, i.e. inhibition causing obesity and activation causing little effects on body weight. This explanation also addresses critiques 3 and 7 below.

- b) **The reviewer’s belief in activation of the melanocortin pathway on reducing body weight**. Although “the redundancy networks in keeping the animal eating are required to maximize survival” has been proposed to explain the current obesity epidemic, this is definitely NOT true for the prevalent view on the role of the melanocortin action. As a matter of fact, this exact criticism raised by the reviewer on the issue of the significance of our study seems to be inconsistent his/her own other criticism raised on the point of the effectiveness of the MC4R agonist in reducing general obesity. If the dogma on redundant pathways to prevent body weight reduction is prevalent in the field, why activation of MC4Rs with agonists is still believed/pursued to reverse general obesity? On the contrary, exactly because the activation of MC4R agonists is still believed to be able to treat general obesity (as clearly suggested by this reviewer), **it is particularly significant to publish our current data showing that activation and gain of function of the melanocortin pathway (at both POMC and MC4R neuron levels) fails to reduce body weight or reverse obesity**. As detailed below in **c-e**, it is also evident that the prevalent belief in the field is that activation of the melanocortin action is able to reduce body weight/reversing obesity. Given these facts, we hope that the reviewer would agree that our current findings on the activation data are significant to the field, as least to initiate a debate on the true function of the melanocortin action. As discussed in our study, our results are significant in that they will help push forward the idea of precision medicine, i.e. obesity therapeutic drugs will be more effective to correct the known defective obesity-causing pathways, for example, Setmelanotide for obese patients with known defects in the melanocortin action (more discussion below).

c) **Prevalent view in the field on the action of melanocortin action in reducing body weight/reversing obesity.** To demonstrate this point, although this view on the melanocortin action in reducing body weight is historically prevalent in the field and has been believed to mediate the function of leptin etc (Cone R, et al., 2005, Nature Neuroscience,; <https://pubmed.ncbi.nlm.nih.gov/15856065/>), we list a few **very recent reviews** on this topic: Hinney A, Nature Rev Endo, 2022, <https://pubmed.ncbi.nlm.nih.gov/35902734/>; Goit RK et al., Eur J. Pharm, 2022: <https://pubmed.ncbi.nlm.nih.gov/35430211/>; Fatima MT et al, Diabetes, Obes. Met., 2022, <https://pubmed.ncbi.nlm.nih.gov/34882941/>; Yeo GSH et al, Mol Met, 2021, <https://pubmed.ncbi.nlm.nih.gov/33684608/>; Vohar MS et al., Eur. J. Pharm, 2022, <https://pubmed.ncbi.nlm.nih.gov/34798121/> and Baldini G et al., J. Endocrinol. 2019, <https://pubmed.ncbi.nlm.nih.gov/30812013/>). For convenience, we have listed pubmed links here so that the reviewer could directly access these links to verify. One common view of these reviews is the belief that activation of the melanocortin action is able to reduce weight and reverse obesity, and can be used therapeutic targets for obesity. For

your convenience, I chose one diagram (Vohar et al, Eur. J. Pharm, 2022) to illustrate this point.

- d) **Prevalent view on gain of function of POMC neurons in reducing body weight/reversing obesity**. Here again we only chose a few **recent** publications on POMC neurons. 1) Qiu J et al. Mol Met 2022: <https://pubmed.ncbi.nlm.nih.gov/36442744/> : Deletion of stromal interaction molecule 1 in POMC-Cre expressing cells proects DIO; 2) Chu G et al., Front Endocrinol 2022: <https://pubmed.ncbi.nlm.nih.gov/36387867/> : showing an increasing POMC level that could mediate the body wieght reducing effect of liver-expressed antimicrobial peptide 2 (LEAP2), a newly discovered antagonist of the growth hormone secretagogue receptor (GHSR); 3) Niraula A et al., Diabetes, 2022: <https://pubmed.ncbi.nlm.nih.gov/36318114/> : better POMC neuron morphology used as explanation for the improved DIO by microglial EP4 deletion. Although it is not in the position for our current study to explain their results or reconcile the data interpretation (POMC-Cre expresses in multiple sites and also target a subset of AgRP neurons etc), these publications clearly demonstrate a prevalent view on the role of POMC neurons in reducing body weight/reversing obesity.
- e) **Prevalent view on gain of function of MC4R neurons in reducing body weight/reversing obesity**. Again we only showed a few **recent** publications on MC4R neurons. 1) Yong Y et al., PNAS, 2021: <https://pubmed.ncbi.nlm.nih.gov/34654741/> : PVH MC4R mediating the body weight reducing effect of eCB 2-arachidonoylglycerol (2-AG); 2) Li MM et al., Neuron, 2019: <https://pubmed.ncbi.nlm.nih.gov/30879785/> : results suggesting MC4R neurons representing one major pathway in preventing obesity; 3) Bruschetta G et al., Mol Met, 2018: <https://pubmed.ncbi.nlm.nih.gov/30352741/> : results showing improving MC4R signaling in PVH MC4R neurons in reducing body weight.

Importantly, for human obesity results, extensive efforts have been made to identify gain-of-function of MC4R in humans, again **demonstrating a strong belief of gain-of-function in reducing weight in the field**. As a result, Lotta LA et al, Cell, 2019: <https://pubmed.ncbi.nlm.nih.gov/31002796/> indeed suggests one gain-of-function MC4R mutation associated with reduced weight in humans. However, there is some controversy with the method used in this paper: Lotta LA et al, Cell, 2021: <https://pubmed.ncbi.nlm.nih.gov/33798435/> and a follow up study with the same mutation in mouse models suggesting a minimal action in body weight: Rojo D et al., Mol Met: <https://pubmed.ncbi.nlm.nih.gov/32916307/> and please see the body weight data of 2 gain-of-function of MC4R models were pasted below to illustrate no body weight effects in mice with these gain-of-function of MC4Rs. In addition, an independent study suggests that gain-of-function of MC4Rs in humans causes no obesity: Wade KH et al., Nat Med, 2021: <https://pubmed.ncbi.nlm.nih.gov/34045736/> . From these publications on both mice and humans, it is clear that it is general belief in the obesity research field that activation or gain of function of MC4Rs is able to reduce body weight/reverse obesity, illustrating the significance of our current findings.

2. The abstract also makes a bold statement about “the largely failed efforts in identifying effective therapeutics toward treating general obesity via activation of the melanocortin action,” when this is simply not true. Setmelanotide has had phenomenal success in altering feeding and body weight. True, while it has not been granted governmental permission to be tested on patients with general obesity, there is very little evidence suggesting it won’t be just as successful as observed in patients with melanocortin mutations. Furthermore, additional compounds targeting this system have been shown to have similar weight-reducing properties in pre-clinical studies.

Responses: We thank the reviewer for the reminder of careful use and citation of previous work in concluding general statements. Regarding this statement, which is directly quoted from a previous review: Fani L et al., *Int J. Obesity*, 2014: <https://pubmed.ncbi.nlm.nih.gov/23774329/>, and the conclusion paragraph was pasted below to demonstrate the similar statement. Although it was published in 2014, it seems to hold true, at least as of today, since Setmelanotide (more discussion below) has not been approved for the general obesity population, i.e. those without gene mutations in the melanocortin pathway. So, in this sense, our statement should be in principle valid.

REVIEW

The melanocortin-4 receptor as target for obesity treatment:
a systematic review of emerging pharmacological
therapeutic options

L Fani^{1,3}, S Bak^{1,3}, P Delhanty², EFC van Rossum² and ELT van den Akker¹

In conclusion, there are currently no effective MC4R agonists for clinical treatment of obese humans. Despite there being no treatment, research for the defects in the pathways of MC4R are emerging, which provide insight into mechanisms that could be used as novel targets for future therapy. From this systematic review, we have come to the conclusion that there can be potential drugs for the treatment of MC4R-mutated and -obese patients in the future, while careful observation of clinical side effects in humans.

The MC4R agonist Setmelanotide has been shown to have anti-obesity effects in humans with known genetic mutations in the melanocortin pathway: Clement KC et al., *Lancet Diabetes Endocrinology*, 2020: <https://pubmed.ncbi.nlm.nih.gov/33137293/>, which is understandable and predicted as this is a type of functional reconstitution therapy, i.e. to supplement what is missing in the system, just as leptin in treating obesity with leptin deficiency. However, as leptin failed to reduce obesity in general obesity population, the effect of Setmelanotide on general obesity

population has been doubtful, which should otherwise have been actively sought for and validated during the last few years since the approval of this drug.

Specifically on this point, a previous animal study: Collet TH et al., Mol Met, 2017: <https://pubmed.ncbi.nlm.nih.gov/29031731/> showed a very mild effect of body weight reduction (data pasted above, please be mindful that the Y axis starts at 20g). A new clinical trial in human Alstrom syndrome severe obesity (BMI>30) with Setmelanotide within 52 weeks produced inclusive results: Haqq AM et al., Lancet Diabetes Endocrinology, 2022: <https://pubmed.ncbi.nlm.nih.gov/36356613/>. Collectively, these data cast a doubt on its effect on general obesity population, most of which are with moderate obesity.

Although it is too early to say whether it will be eventually approved in general obesity population, this drug approval status should not affect the significance of our current findings. On the contrary, **the uncertainty of the Setmelanotide effect on general obesity clearly demonstrates the special significance of our findings on activation of the melanocortin action causing no reduction in body weight or reversal of obesity, indicating a need to differentiate the effect of activation of the melanocortin action in treating obesity caused by specific genetic mutations from treating general obesity.**

3. *Kir2.1-inhibition of POMC neurons recapitulates both the short and long-term hyperphagia and weight gain previously reported with POMC inhibition/ablation so this does not add much to what is known.*

Responses: The importance of inhibition data on POMC neurons has been addressed in 1) above.

4. *Unless there is a difference observed between males and females, I'd suggest combining the sex data into one graph instead of two separate ones for each measurement.*

Responses: We thank the reviewer for this suggestion but would like to respectfully point out that it may be counter-productive to combine data from both sexes, especially for this study, in which we intend to make claims that manipulations will not cause difference in metabolism. For example, given the known differences in females (smaller numbers in body weight and others) and males (bigger numbers in body weight and others), combination of these data will cause a much greater variation in each group, which will cause a concern that no differences observed are simply due to a large variation within the datasets. We hope that the reviewer would agree that the current way of data presentation in each sex will be more robust and convincing.

5. *One of the biggest weaknesses of the manuscript is the lack of verification of overexpressed proteins. While immunohistochemistry is performed (but not quantified; more on this below), additional assays would be required to ensure proper functionality beyond what is shown. For*

example, n=4 to show that at a very specific time point (2 hours) that re-feeding is lower confirms functionality of alpha-MSH is weak.

Responses: We used both immunohistochemistry and functional verification for each of gain of function model to verify the successful and functional gain of function. we have now provided quantification data to support the difference in expression. Specifically, for the gain-of-function of alpha-MSH model, although we only have n=4, the data are very consistent and convincingly demonstrated the expected effect on feeding reduction. Thus, it is in our opinion that this dataset is sufficient to show a functional gain of function of overexpressed alpha-MSH.

6. The authors often use representative images to highlight a point of text but without proper quantification, it is hard to decipher and becomes hard to believe. For example, the alpha-MSH staining in Figure 3b on the non-injection side appears extremely low at baseline when the authors want to demonstrate that injection of the construct leads to robust expression in comparison but then baseline alpha-MSH appears high in the control injected with beta-endorphin in Figure 4c.

Responses: We thank the reviewer for pointing out the potential concern from various fluorescence differences in our illustrated pictures. After careful examination, we think the difference in the level of control side between the two figures may be due to an auto-adjustment of confocal, i.e. when the contralateral side has a higher level intensity of signal, it will automatically reduce the setting to prevent over-exposure. However, it is in our view these differences between study subjects does not matter in any way in this case because **we only compare signals within the same subjects**, in other words, we will not compare expression levels between animals but instead only compared the signals between the injected side and non-injected side within the same animal. Since the picture was taken from a particular brain section with the same setting and the same anatomical location, we view this comparison will avoid potential biases introduced during the picture-taking procedure (exposure time and intensity etc) or potential different brain locations, and therefore should be a more rigorous way to demonstrate the changes in the expression level.

7. Similar to the POMC experiments, prior work has demonstrated that acute inhibition of MC4R PVN neurons or specific knockout of Mc4r drive obesity.

Responses: Again the necessity of including the inhibition data of MC4R neurons has been addressed in **1)** above.

In summary, we believe we have effectively addressed reviewers' concerns, especially we have laid out factual evidence demonstrating a prevalent belief in the field (including reviewer 2) that activation of the melanocortin action is able to effectively reduce body weight/reverse obesity. Thus, our current findings are significant to the field as it will modify, or at minimum initiate necessary debate on, the current canonical view on the bidirectional action of the melanocortin pathway in body weight regulation.

REVIEWER COMMENTS

Reviewer #1 (Remarks to the Author):

My concerns are mostly addressed. However, the authors should revise the statement "the melanocortin action drives one-directional obesity development" in the abstract. The manuscript does not present any evidence showing that activation of the melanocortin pathway causes obesity. The authors should consider replacing "the largely failed efforts" with "the difficulty" in the abstract, which should address one of reviewer 2's concerns.

Congratulations to the authors for this interesting and well-controlled study! I think the finding that enhancing melanocortin signaling neither prevents obesity development nor reverses obesity is novel and important.

Baoji Xu

Reviewer #2 (Remarks to the Author):

In this revision Li et al reinforce observations that inhibition of POMC and PVH MC4R cause massive obesity and demonstrate here that unexpectedly, chronic activation or overexpression of anorexigenic peptides fails to alter body weight.

I didn't mean to downplay the significance of the findings in my previous review, only to highlight that the animal brain has evolved in a manner that ensures proper levels of energy stores to prevent starvation. Thus, I was just stating that it would be unlikely that any one pathway could reverse body weight gain and I think the authors would agree. However, the findings of this manuscript are still important in showing that melanocortin action is biased toward weight gain and not loss especially given the wealth of obesity drugs on the horizon, some of which have been designed to specifically target this system and that have had far less success than those targeting other signaling/receptor systems.

Major comments

- While some quantification efforts have been made, there are still several places in the manuscript where a statement is made in the text and this is only supported by a representative image. If such claims are made, then it needs to be reinforced with quantifiable data and that can be compared statistically to ensure reproducibility (ie. number of cells, colocalization, intensity of fluorescence), even if it may seem obvious in the selected pictograph shown to the reader. This applies to Figures 2d-e, 3c, 4c, 5a-b, 6a, 7a, Supplementary Figures 4a, 5a and 7a.

Minor comments below:

- For symmetry, it would be insightful if the authors have data demonstrating that chronic activation of PVH MC4R neurons does not reverse obesity like they show so nicely for POMC neurons
- When describing the effects on the PVH MC4R chronic activation and inhibition studies, I would suggest switching the adverb “Strikingly” to describe the results of the chronic activation study (previously unknown and assumed to promote weight loss) from where it is currently in describing the obesity brought on by chronic inhibition (which would be highly expected given knockouts and perturbations of the receptor and neurons, respectively).
- As far as I know there are very few models whereby chronic activation or inhibition lead to reduced body weight but the Palmiter lab demonstrated that acute AgRP ablation leads to starvation and severe weight loss. Perhaps this model should be highlighted more in the Discussion. There is a quick reference to it with regards to it reducing feeding but to put the whole melanocortin circuitry in perspective it would be useful to have this all together.
- Out of curiosity, would it be possible to overexpress these proteins (POMC, alpha-MSH, MC4R) in cell types that don’t normally express them? For example, I wonder what the effects would be of overexpressing MC4Rs throughout the entire PVH or in other regions shown to reduce food intake. Or for that matter overexpression of POMC or alpha-MSH in AgRP neurons.

We would like to thank both reviewers for their critical reviews, which really helped improve the manuscript. Here we provide point-by-point responses to reviewers' concerns. All changes made in the text are labelled in red.

Reviewer 1:

My concerns are mostly addressed. However, the authors should revise the statement "the melanocortin action drives one-directional obesity development" in the abstract. The manuscript does not present any evidence showing that activation of the melanocortin pathway causes obesity. The authors should consider replacing "the largely failed efforts" with "the difficulty" in the abstract, which should address one of reviewer 2's concerns.

Congratulations to the authors for this interesting and well-controlled study! I think the finding that enhancing melanocortin signaling neither prevents obesity development nor reverses obesity is novel and important.

Response: We would like to thank the reviewer for the enthusiasm on our results and appreciate the suggestion on the choice of wording. We have made the changes in the text. Specifically, we have replaced "the melanocortin action drives one-directional obesity development" with "Changes in the melanocortin action cause one-directional obesity development".

Reviewer 2:

Point 1: In this revision Li et al reinforce observations that inhibition of POMC and PVH MC4R cause massive obesity and demonstrate here that unexpectedly, chronic activation or overexpression of anorexigenic peptides fails to alter body weight.

I didn't mean to downplay the significance of the findings in my previous review, only to highlight that the animal brain has evolved in a manner that ensures proper levels of energy stores to prevent starvation. Thus, I was just stating that it would be unlikely that any one pathway could reverse body weight gain and I think the authors would agree. However, the findings of this manuscript are still important in showing that melanocortin action is biased toward weight gain and not loss especially given the wealth of obesity drugs on the horizon, some of which have been designed to specifically target this system and that have had far less success than those targeting other signaling/receptor systems.

Response: We would like to thank the reviewer for the clarifications and agreeing on the significance of this work.

Point 2. While some quantification efforts have been made, there are still several places in the manuscript where a statement is made in the text and this is only supported by a representative image. If such claims are made, then it needs to be reinforced with quantifiable data and that can be compared statistically to ensure reproducibility (ie. number of cells, colocalization, intensity of fluorescence), even if it may seem obvious in the selected pictograph shown to the reader. This applies to Figures 2d-e, 3c, 4c, 5a-b, 6a, 7a, Supplementary Figures 4a, 5a and 7a.

Response: We thank the reviewer for pointing out the importance of additional quantitative data. As requested, we have now provided these data in new Figs 2f and 2g (for pictures shown in Figs. 2d-e), Fig. 3d (for pictures in Fig. 3c), Supplementary Fig. 5b (for pictures in Fig. 4c), Supplementary Figs. 6b and 6b (for pictures in Fig. 5a-b), Fig. 6b (for pictures in Fig. 6a),

Supplementary Fig. 8a (for pictures in Fig. 7a), Supplementary 4j (for pictures in Supplementary Fig. 4a), Supplementary Fig. 5c (for pictures in Supplementary Fig. 5a); and Supplementary 7b (for pictures in Supplementary Fig. 7a). These quantitative data substantiate the animal models used for the experiments.

Point 3. For symmetry, it would be insightful if the authors have data demonstrating that chronic activation of PVH MC4R neurons does not reverse obesity like they show so nicely for POMC neurons.

Response: We agree with the reviewer on the effect related to HFD-induced obesity. We have now included new data on chronic activation of PVH MC4R neurons in HFD-induced obesity (new Fig. 6d). In addition, we have also included body weight with HFD feeding in Supplementary Fig. 8f, in which mice with chronic activation of both POMC and PVH MC4R neurons to achieve maximal activation of the melanocortin pathway. In both cases, no differences were observed in body weight with mice fed HFD, suggesting that chronic activation PVH MC4Rs or the melanocortin pathway is unable to reduce obesity even under HFD-induced obesogenic conditions.

Point 4. When describing the effects on the PVH MC4R chronic activation and inhibition studies, I would suggest switching the adverb “Strikingly” to describe the results of the chronic activation study (previously unknown and assumed to promote weight loss) from where it is currently in describing the obesity brought on by chronic inhibition (which would be highly expected given knockouts and perturbations of the receptor and neurons, respectively).

Response: We have removed the word “Strikingly”.

Point 5. As far as I know there are very few models whereby chronic activation or inhibition lead to reduced body weight but the Palmiter lab demonstrated that acute AgRP ablation leads to starvation and severe weight loss. Perhaps this model should be highlighted more in the Discussion. There is a quick reference to it with regards to it reducing feeding but to put the whole melanocortin circuitry in perspective it would be useful to have this all together.

Response: We have expanded the discussion and added “Under this context, The starvation phenotype by *i.m.* toxin induced AgRP neuron lesion in adult mice appears to be surprising. Since AgRP is also expressed in the adrenal gland⁵⁶, more studies with specific AgRP neuron lesion in the brain is warranted to specifically address this issue.” in the discussion.

Point 6. Out of curiosity, would it be possible to overexpress these proteins (POMC, alpha-MSH, MC4R) in cell types that don’t normally express them? For example, I wonder what the effects would be of overexpressing MC4Rs throughout the entire PVH or in other regions shown to reduce food intake. Or for that matter overexpression of POMC or alpha-MSH in AgRP neurons.

Response: We thank the reviewer for the suggestions. However, we want to respectfully suggest that the results of the proposed experiments will not have any bearing on the conclusion drawn in the current manuscript. The proposed animal models may not be able to mimic the condition of activation of the endogenous melanocortin pathway.

REVIEWERS' COMMENTS

Reviewer #2 (Remarks to the Author):

The authors have addressed my previous comments.